# BiGTA-Net: A Hybrid Deep Learning-Based Electrical Energy Forecasting Model for Building Energy Management Systems

Dayeong So [1], Jinyeong Oh [2], Insu Jeon [3], Jihoon Moon [1,2,3,*], Miyoung Lee [4] and Seungmin Rho [5,*]

[1] Department of ICT Convergence, Soonchunhyang University, Asan 31538, Republic of Korea; sodayeong@sch.ac.kr
[2] Department of AI and Big Data, Soonchunhyang University, Asan 31538, Republic of Korea; wlsdud8261@sch.ac.kr
[3] Department of Medical Science, Soonchunhyang University, Asan 31538, Republic of Korea; jis601@sch.ac.kr
[4] Department of Software, Sejong University, Seoul 05006, Republic of Korea; miylee@sejong.ac.kr
[5] Department of Industrial Security, Chung-Ang University, Seoul 06974, Republic of Korea
* Correspondence: jmoon22@sch.ac.kr (J.M.); smrho@cau.ac.kr (S.R.)

**Abstract:** The growth of urban areas and the management of energy resources highlight the need for precise short-term load forecasting (STLF) in energy management systems to improve economic gains and reduce peak energy usage. Traditional deep learning models for STLF present challenges in addressing these demands efficiently due to their limitations in modeling complex temporal dependencies and processing large amounts of data. This study presents a groundbreaking hybrid deep learning model, BiGTA-net, which integrates a bi-directional gated recurrent unit (Bi-GRU), a temporal convolutional network (TCN), and an attention mechanism. Designed explicitly for day-ahead 24-point multistep-ahead building electricity consumption forecasting, BiGTA-net undergoes rigorous testing against diverse neural networks and activation functions. Its performance is marked by the lowest mean absolute percentage error (MAPE) of 5.37 and a root mean squared error (RMSE) of 171.3 on an educational building dataset. Furthermore, it exhibits flexibility and competitive accuracy on the Appliances Energy Prediction (AEP) dataset. Compared to traditional deep learning models, BiGTA-net reports a remarkable average improvement of approximately 36.9% in MAPE. This advancement emphasizes the model's significant contribution to energy management and load forecasting, accentuating the efficacy of the proposed hybrid approach in power system optimizations and smart city energy enhancements.

**Keywords:** energy management system; short-term load forecasting; building energy forecasting; hybrid deep learning model; bi-directional gated recurrent unit; temporal convolutional network

## 1. Introduction

The rapid influx of populations into urban areas presents many challenges, ranging from resource constraints to heightened traffic and escalating greenhouse gas (GHG) emissions [1]. Many cities globally are transitioning towards 'smart cities' to handle these multifaceted urban issues efficiently [2]. At its core, a smart city aims to enhance its inhabitants' efficiency, safety, and living standards [3]. For example, smart cities tackle GHG emissions by reducing traffic congestion, optimizing energy usage, and incorporating alternatives, such as electric vehicles, energy storage systems (ESSs), and sustainable energy sources [4]. A significant portion of urban GHG emissions is attributed to building electricity consumption, which powers essential systems and amenities such as heating, domestic hot water (DHW), ventilation, lighting, and various electronic appliances [5]. Thus, advancing energy efficiency in urban buildings, especially through the integration of energy storage and renewable energy sources, is paramount. Recognizing this, many smart city designs have embraced integrated systems such as building energy management systems (BEMSs) to boost the energy efficiency of existing infrastructure [6].

A BEMS is a technology-driven tool that harnesses the capabilities of the Internet of Things (IoT) [7] and big data analytics [8] to specifically regulate and monitor building electricity consumption. Electricity accounts for a substantial portion of a building's energy profile, powering everything from lighting and heating to advanced electronic appliances. One of the cardinal functions of a BEMS is short-term load forecasting (STLF) for electricity [9]. Accurate STLF is essential as it enables facilities to trade surplus electricity, foster economic benefits, and precisely manage power loads, thereby preventing blackouts by moderating peak electrical demands [10]. However, mastering STLF for electricity consumption is an intricate task. This is because buildings exhibit diverse and complex electricity consumption patterns with a non-linear relationship with several external factors such as weather, occupancy, and time of day [11]. Additionally, the inherent noise in electricity consumption data further muddles the forecasting process, making accurate predictions challenging [12]. Given these complexities, many researchers have turned to artificial intelligence (AI) as a promising approach for building electricity consumption forecasting. AI techniques excel in deciphering the recent trends in electricity consumption and processing the intricate, non-linear interactions between various influencing factors and electricity demand [13].

Recent research underscores the intricate dynamics governing building energy performance. Several factors, both internal, such as building orientation, and external, such as climatic changes, play pivotal roles. These complexities can be unraveled through mathematical modeling, leading to the formulation of more accurate regression models [14]. Delving into the digital realm, machine learning (ML) stands out as a potent tool. With the support of vast datasets and advanced computing, ML offers groundbreaking solutions for predicting energy demands [15]. Its potential is evident throughout a building's lifecycle, impacting both its design and operation phases. However, the journey to its broad acceptance presents numerous challenges. Two notable hurdles include the necessity for extensive labeled data and concerns regarding model transferability. In response to these challenges, innovative strategies have emerged. One notable approach is the introduction of easy-to-install forecast control systems designed for heating. These systems prove especially beneficial for older structures that necessitate detailed documentation [16]. Such systems not only exemplify technical advancements but also adapt to diverse inputs, considering factors from weather conditions to occupant preferences, ensuring an optimal balance between energy efficiency and occupant comfort.

Building on the promise of ML, as highlighted in recent research, traditional AI techniques, including ML [13] and deep learning (DL) [17], have indeed been extensively employed to develop forecasting models. Several ML approaches have been explored, showcasing innovative methodologies to predict hourly or peak energy consumption. Granderson et al. [18] focused on the versatility of regression models in predicting hourly energy consumption. By emphasizing its applicability in STLF, their model showcased its potential in the broader realm of energy management. Huang et al. [19] presented a multivariate empirical mode decomposition-based model that harmoniously integrated particle swarm optimization (PSO) and support vector regression for day-ahead peak load forecasting. Li et al. [20] pioneered a data-driven strategy for STLF by integrating cluster analysis, Cubist regression models, and PSO, presenting an innovative approach that balanced multiple techniques for improved forecasting. Moon et al. [21] introduced the ranger-based online learning approach, called RABOLA, a testament to adaptive forecasting specially tailored for buildings with intricate power consumption patterns. This model prioritized real-time, multistep-ahead forecasting, demonstrating its potential in dynamic environments.

While traditional ML methods have significantly advanced energy forecasting, the advent of DL, especially convolutional neural networks (CNNs) and recurrent neural networks (RNNs), has opened new horizons. These neural networks have set a precedent for understanding intricate data patterns, paving the way for more sophisticated forecasting methodologies [22]. Compared to traditional ML and mathematical methods, these models

stand out due to their learning capabilities and generalization ability [13]. Understanding the characteristics of building electricity consumption data, including time-based [23] and spatial features [24], is essential for efficiently applying these DL models. Despite these sophisticated techniques, the models often deliver unreliable, low forecasts [25]. They need help with several challenges, such as issues related to short-term memory, overfitting, learning from scratch, and understanding complex variable correlations. Some researchers have investigated hybrid models to overcome these challenges, as single models frequently have difficulties learning time-based and spatial features simultaneously [13].

Building upon the aforementioned advances in forecasting techniques, a multitude of studies, including those mentioned above, have delved into the realm of STLF. These collective efforts spanning several years are comprehensively summarized in Table 1. Taking a leaf from hybrid model designs, Aksan et al. [26] introduced models that combined variational mode decomposition (VMD) with DL models, such as CNN and RNNs. Their models, VMD-CNN-long short-term memory (LSTM) and VMD-CNN-gated recurrent unit (GRU), showcased versatility, adeptly managing seasonal and daily energy consumption variations. Wang et al. [27], in their recent endeavors, proposed a wavelet transform neural network that uniquely integrates time and frequency-domain information for load forecasting. Their model leveraged three cutting-edge wavelet transform techniques, encompassing VMD, empirical mode decomposition (EMD), and empirical wavelet transform (EWT), presenting a comprehensive approach to forecasting. Zhang et al. [28] emphasized the indispensable role of STLF in modern power systems. They introduced a hybrid model that combined EWT with bidirectional LSTM. Moreover, their model integrated the Bayesian hyperparameter optimization algorithm, refining the forecasting process. Saoud et al. [29] ventured into wind speed forecasting and introduced a model that amalgamated the stationary wavelet transform with quaternion-valued neural networks, marking a significant stride in renewable energy forecasting.

**Table 1.** Comparative analysis of previous studies and the current research concerning short-term load forecasting.

| Researchers | Model Used | Addresses Long-Term Dependencies | Manages Varying Input Sequence Lengths | Weighs Different Features |
|---|---|---|---|---|
| Granderson et al. [18] | Regression model | No | Partially | No |
| Huang et al. [19] | MEMD-PSO-SVR | Partially | Partially | Partially |
| Li et al. [20] | Cluster analysis, Cubist regression models, PSO | No | No | Partially |
| Moon et al. [21] | RABOLA | Partially | Partially | Partially |
| Aksan et al. [26] | VMD-CNN-GRU and LSTM | Yes | Partially | Partially |
| Wang et al. [27] | Wavelet transformer, LSTM | Yes | No | Yes |
| Zhang et al. [28] | Bi-LSTM, BHO, EWT | Yes | Partially | Yes |
| Kim et al. [30] | RNN, 1D-CNN | Partially | Yes | Partially |
| Jung et al. [31] | Attention-GRU | Yes | No | Partially |
| Zhu et al. [32] | LSTM based dual-attention model | Partially | Partially | Partially |
| Liao et al. [33] | LSTM, TPA mechanism | Yes | No | Yes |
| BiGTA-net | Bi-GRU, TCN, attention mechanism | Yes | Yes | Yes |

Kim et al. [30] seamlessly merged the strengths of RNN and one-dimensional (1D)-CNN for STLF, targeting the refinement of prediction accuracy. They adjusted the hidden state vector values to suit closely better-occurring prediction times, showing a marked evolution in prediction approaches. Jung et al. [31] delved into attention mechanisms (Att) with their Att-GRU model for STLF. Their approach was particularly noteworthy for adeptly managing sudden shifts in power consumption patterns. Zhu et al. [32] showcased

an advanced LSTM-based dual-attention model, meticulously considering the myriad of influencing factors and the effects of time nodes on STLF. Liao et al. [33], with their innovative fusion of LSTM and a time pattern attention mechanism, augmented STLF methodologies, emphasizing feature extraction and model versatility. By incorporating external factors, their comprehensive approach improved feature extraction and demonstrated superior performance compared to existing methods. While effective, their model should have capitalized on the strengths of hybrid DL models, such as GRU and temporal convolutional network (TCN), which could be used to handle both long-term dependencies and varying input sequence lengths [34].

BiGTA-net is introduced as a novel hybrid DL model that seamlessly integrates the strengths of a bi-directional gated recurrent unit (Bi-GRU), a temporal convolutional network (TCN), and an attention mechanism. These components collectively address the persistent challenges encountered in STLF. The conventional DL models sometimes require assistance in dealing with intricate nonlinear dependencies. However, the amalgamation within the proposed model represents an innovative approach for capturing long-term data dependencies and effectively handling diverse input sequences. Moreover, the incorporation of the attention mechanism within BiGTA-net optimizes the weighting of features, thereby enhancing predictive accuracy. This research establishes its unique contribution within the energy management and load forecasting domains, which can be attributed to the following key contributions:

- BiGTA-net emerges as a pioneering hybrid DL model designed to enhance day-ahead forecasting within power system operation, prioritizing accuracy.
- The experimental framework employed for testing BiGTA-net's capabilities is strategically devised, showcasing its adaptability and resilience across different models and configurations.
- Utilizing data sourced from a range of building types, the approach employed in this study establishes the widespread applicability and adaptability of BiGTA-net across diverse consumption scenarios.

The structure of this paper is outlined as follows: Section 2 elaborates on the configuration of input variables that are crucial to the STLF model and discusses the proposed hybrid deep learning model, BiGTA-net. In Section 3, the performance of the model is thoroughly examined through extensive experimentation. Finally, Section 4 encapsulates the findings and provides an overview of the study.

## 2. Materials and Methods

This section provides an in-depth exploration of the meticulous processes utilized to structure the datasets, configure the models, and assess their performance. Serving as an initial reference, Figure 1 displays a block schema that visually encapsulates the progression of the approach from raw datasets to performance evaluation. This schematic illustration is essential in providing readers with a comprehensive perspective of the methodological steps, emphasizing critical inputs, outputs, and incorporated innovations.

### 2.1. Data Preprocessing

This section explains the procedure undertaken to identify crucial input variables necessary for shaping the STLF model. Central to this study is the forecast of the day-ahead hourly electricity load. This forecasting holds immense significance, primarily due to its role as a foundational element in the planning and optimization of power system operations for the upcoming day [35]. These forecasts contribute to the following aspects:

- Demand Response: An approach centered on adjusting electricity consumption patterns rather than altering the power supply. This method ensures the power system can cater to fluctuating demands without overextending its resources.
- ESS Scheduling: This entails critical decisions on when to conserve energy in storage systems and when to discharge it. Effective scheduling ensures optimal stored energy utilization, aligning with predicted demand peaks and troughs.

- Renewable Electricity Production: Anticipating the forthcoming electricity demand facilitates strategic planning for harnessing renewable sources. It ensures renewable sources are optimally utilized, considering their intermittent nature.

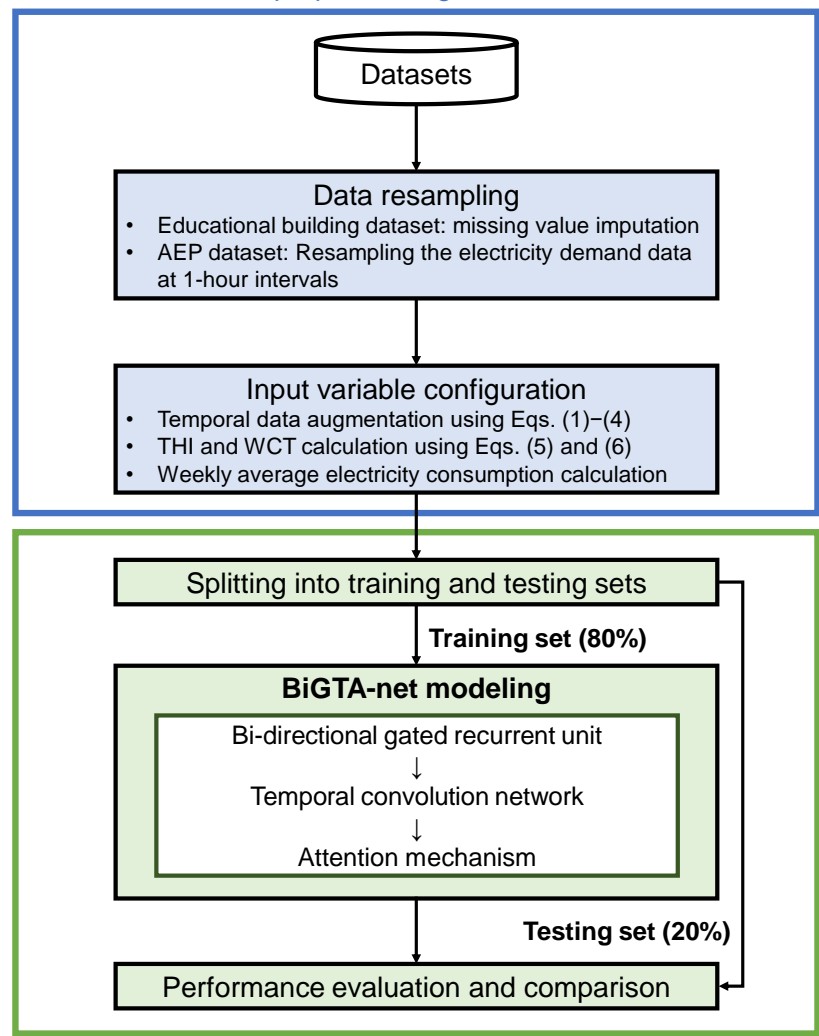

**Figure 1.** Schematic flow of data preprocessing and BiGTA-net modeling.

The study explores two distinct datasets that represent divergent building types, contributing to a comprehensive understanding of power consumption patterns and enhancing the formulation of the model. The first dataset originates from Sejong University, exemplifying educational institutions [36]. In contrast, the Appliances Energy Prediction (AEP) dataset represents residential buildings [37]. The objective was to enhance the precision of the STLF model by incorporating insights from these datasets, ensuring its adaptability to various electricity consumption scenarios.

Sejong University employed the Power Planner tool, which generates electricity usage statistics, to optimize electricity consumption. These statistics include predicted bills, electricity consumption, and load pattern analysis. Five years' worth of hourly electricity consumption data, spanning from March 2015 to February 2021, were compiled using this tool. From the collected dataset, approximately 0.006% of time points (equivalent to 275 instances) contained missing values, which were imputed based on prior research on handling missing electricity consumption data. Conversely, the publicly available AEP dataset provides residential electricity consumption data at 5 min intervals. To align with

the study's objective of predicting day-ahead hourly electricity consumption, this dataset was resampled at 1 h intervals.

Details of the building electricity consumption, including statistical analysis, data collection periods, and building locations, are presented in Table 2, while Figure 2 illustrates the electricity consumption distribution through a histogram. Figures 3 and 4 illustrate boxplots representing the hourly electricity consumption. Figure 3 presents the consumption data segmented by hours for two datasets: the educational building dataset (Figure 3a) and the AEP dataset (Figure 3b). Similarly, Figure 4 provides boxplots of the same consumption data, which is segmented by days of the week, again for the educational building dataset (Figure 4a) and the AEP dataset (Figure 4b). The minimum and maximum values in Table 2 are omitted due to university privacy concerns. Analysis of Figure 3 revealed a clear distinction in electricity consumption during work hours and non-work hours for both datasets. While the educational building dataset showed a noticeable variation in electricity consumption between weekdays and weekends, the AEP dataset needed to show such a clear distinction.

**Table 2.** Building electricity consumption dataset information.

| Statistics | Educational Building (Unit: kWh) [36] | Appliances Energy Prediction (Unit: Wh) [37] |
|---|---|---|
| Number of samples | 43,848 | 3289 |
| Mean | 2183.70 | 586.18 |
| Standard deviation | 756.41 | 488.98 |
| Median | 1950.84 | 380 |
| Trimmed mean | 2104.24 | 476.53 |
| Median absolute deviation | 708.80 | 163.09 |
| Range | 3793.32 | 3830 |
| Skew | 0.79 | 2.41 |
| Kurtosis | −0.39 | 6.49 |
| Standard error | 3.61 | 8.53 |
| Data collection period | 1 March 2015−29 February 2020 | 11 January 2016−27 May 2016 |
| Building location | Seoul, Republic of Korea | Stambruges, Belgium |
| Public access | No | Yes |

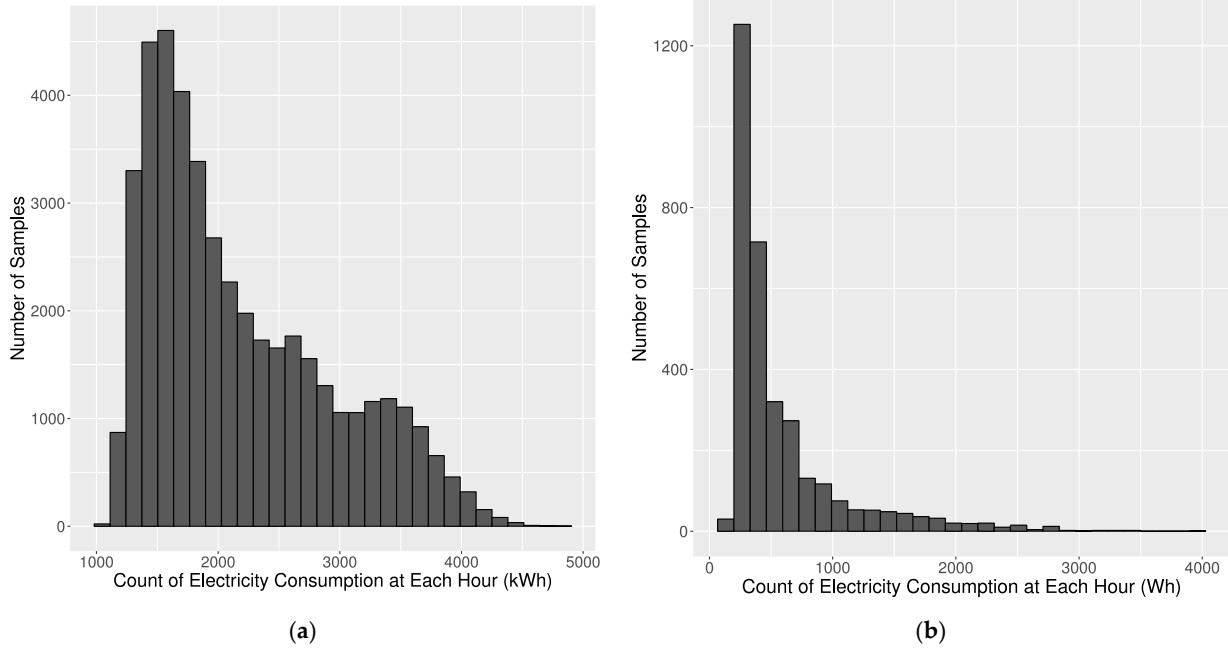

**Figure 2.** Distribution of hourly electricity consumption of a building. (**a**) Educational building dataset; (**b**) Appliances Energy Prediction dataset.

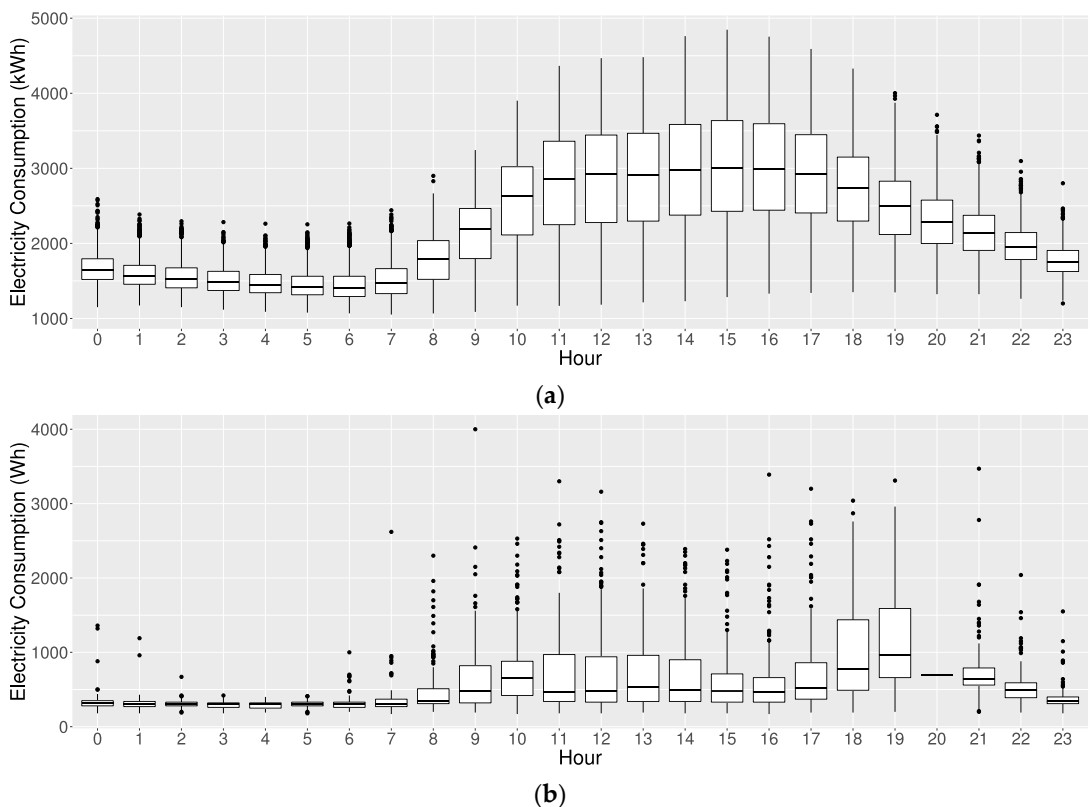

**Figure 3.** Boxplots for hourly electricity consumption by hours. (**a**) Educational building dataset; (**b**) Appliances Energy Prediction dataset.

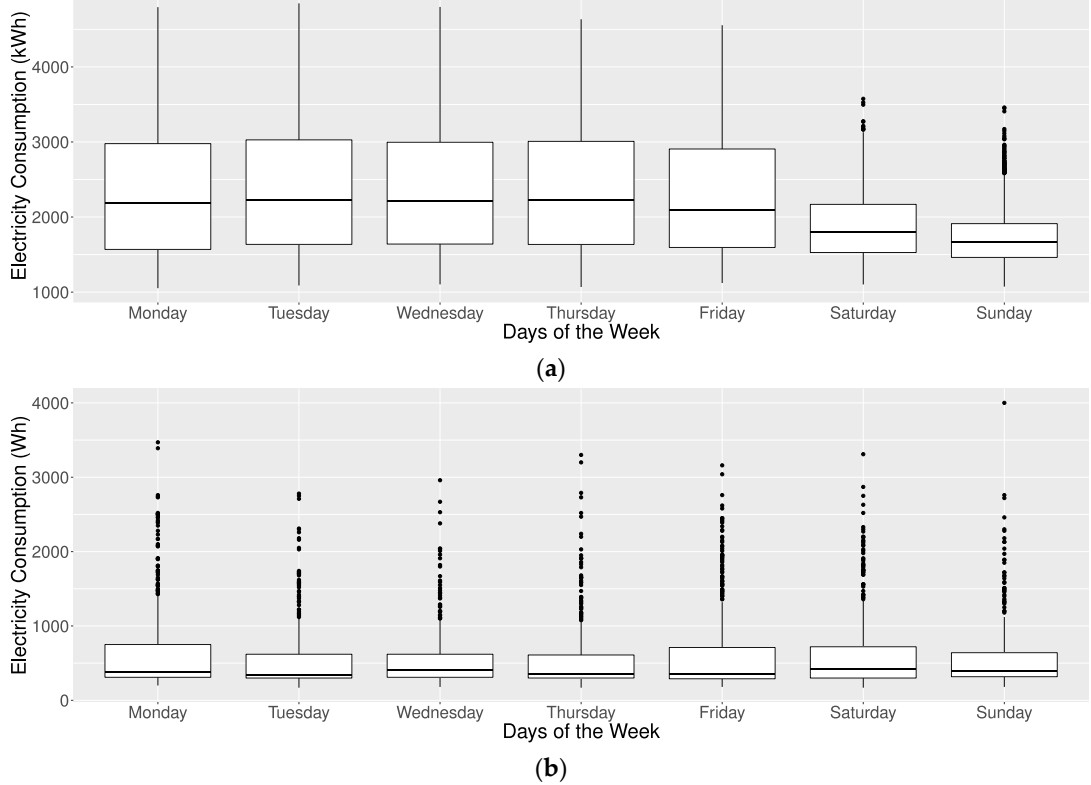

**Figure 4.** Boxplots for hourly electricity consumption by days of the week. (**a**) Educational building dataset; (**b**) Appliances Energy Prediction dataset.

### 2.1.1. Timestamp Information

The study considered a spectrum of external and internal factors in determining the input variables. Among the external factors, timestamps and weather details held significance. These timestamp details encompass the month, hour, day of the week, and holiday indicators. Such details are crucial as they elucidate diverse electricity consumption patterns within buildings. For example, hourly electricity consumption can vary based on customary working hours, mealtime tendencies, and other factors. Similarly, distinct days of the week and holiday indicators can provide insights into contrasting consumption patterns, particularly when contrasting workdays with weekends.

A significant challenge emerges when considering time-related data: the disparity in representing cyclical time data. Specifically, within the hourly context, the difference between 11 p.m. and midnight, though consecutive hours, is illustrated as a substantial gap of 23 units. To address such disparities and effectively capture the cyclic essence of these variables with their inherent sequential structure, a two-dimensional projection was utilized. Equations (1) through (4) were employed to transition from representing these categorical variables in one-dimensional space to depicting them as continuous variables in two-dimensional space [30]:

$$\text{Hour}_x = \sin(360°/24 \times \text{Hour}), \tag{1}$$

$$\text{Hour}_y = \cos(360°/24 \times \text{Hour}). \tag{2}$$

For the day of the week (DOTW) component, considering the ISO 8601 standard where Monday is denoted as one and Sunday as seven, a similar challenge emerges, with a numerical gap of six between Sunday and Monday. This numerical gap can be addressed with the following equations:

$$\text{DOTW}_x = \sin(360°/7 \times \text{DOTW}), \tag{3}$$

$$\text{DOTW}_y = \cos(360°/7 \times \text{DOTW}), \tag{4}$$

here the x and y subscripts in Equations (1) to (4) indicate the two-dimensional coordinates to represent the cyclical nature of hours and days of the week. The transformation to a two-dimensional space allows for a more natural representation of cyclical time data, reducing potential discontinuities.

Beyond these considerations, the analysis also encompassed the integration of holiday indicators [36]. These indicators, denoting weekends and national holidays, were represented as binary variables: '1' indicated a date falling on either a holiday or a weekend, while '0' indicated a typical weekday. By incorporating these indicators, the aim was to account for the evident influence of holidays and weekends on electricity consumption patterns. Notably, the month within a year significantly affects these patterns. However, due to constraints posed by the AEP dataset, which provides data for only a single year, the incorporation of monthly variations was not feasible. As a result, monthly data were not included in the analysis for the AEP dataset.

### 2.1.2. Climate Data

Climate conditions exert a notable influence on STLF, primarily attributed to their integral role within the operational dynamics of high-energy-consuming devices. This influence extends to heating and cooling systems, whose operational patterns align closely with fluctuations in weather conditions [38]. The AEP dataset encompasses six distinct weather variables: temperature, humidity, wind velocity, atmospheric pressure, visibility, and dew point. Conversely, the Korea Meteorological Administration (KMA) offers a comprehensive collection of weather forecast data for each region in South Korea. These data include a range of variables, such as climate observations, forecasts for rainfall likelihood and quantity, peak and trough daily temperatures, wind metrics, and humidity levels [39]. To heighten the real-world applicability of the method, the primary input variables were se-

lectively chosen as temperature, humidity, and wind velocity. This selection was motivated by two factors: firstly, these variables are present both in the AEP dataset and in KMA's forecasts. Secondly, their well-documented strong correlation with power consumption patterns supports their significance [30].

The data reservoir was populated through the automated synoptic observing system of the Korea Meteorological Administration (KMA), maintained by the Seoul Meteorological Observatory. This observatory is located within a mere 10 km of the Sejong University campus. The objective was to contextualize the climatic variables with the environmental conditions of the university's academic buildings. To bridge the gap between raw climatic data and its tangible influence on electricity consumption—the human perceptual experience of temperature fluctuations—two distinct indices were extrapolated. The temperature–humidity index (THI) [40], colloquially known as the discomfort index, provides insights into the perceived discomfort caused by the summer heat, thereby influencing the use of cooling systems. Conversely, the wind chill temperature (WCT) [41] encapsulates the chilling effect of winter weather, often prompting the activation of heating appliances. These perceptual aspects are formulated mathematically in Equations (5) and (6), respectively, where Temp, Humi, and WS represent temperature, humidity, and wind speed.

$$\text{THI} = (1.8 \times \text{Temp} + 32) - [(0.55 - 0.0055 \times \text{Humi}) \times (1.8 \times \text{Temp} - 26)]. \tag{5}$$

Drawing from the feedback loop between temperature, humidity, and the human body's thermoregulation, Equation (5) for THI has been crafted. Its constants—1.8, 32, 0.55, 0.0055, and 26—are the outcome of rigorous empirical studies that evaluated human discomfort across a spectrum of temperature and humidity gradients [40].

$$\text{WCT} = 13.12 + 0.6215 \times \text{Temp} - 11.37 \times \text{WS}^{0.16} + 0.3965 \times \text{Temp} \times \text{WS}^{0.16}. \tag{6}$$

The derivation of Equation (6) for the wind chill temperature (WCT) is grounded in a model that seeks to quantify the perceived decrease in ambient temperature due to wind speed, particularly in colder regions. The constants incorporated within the equation—13.12, 0.6215, 11.37, and 0.3965—as well as the exponent 0.16 trace their origins to comprehensive field experiments conducted across various weather conditions. These experiments were designed to establish a comprehensive model for human tactile perception of cold induced by wind [41]. Taking these considerations into account, the analysis encompassed a set of ten external determinants that were carefully selected as input variables for the model's training process.

### 2.1.3. Past Power Consumption

Past power consumption data were treated as internal factors, as they capture recent patterns in electricity usage [31]. Data from the same time point one day and one week prior were utilized. Power consumption data from the preceding day could provide insight into the most recent hourly trends, while power consumption data from the preceding week could capture the most recent weekly patterns [21]. Given the potential variation in power usage patterns between regular days and holidays, holiday indicators were also integrated for both types of power consumption [36].

Furthermore, an innovative inclusion was made of a past electricity usage value as an internal factor, effectively capturing the trend in electricity consumption leading up to the prediction time point over a span of one week [36]. To achieve this, two distinct scenarios, illustrated in Figure 5, were considered. In the first scenario, if the prediction time point fell on a regular day, the mean electricity consumption of the preceding seven regular days was computed. In the second scenario, if the prediction time point corresponded to a holiday, the average electricity consumption of the preceding seven holidays was calculated. As a result, five internal factors were incorporated for model training, and a comprehensive list of all input variables and their respective details can be found in Table 3.

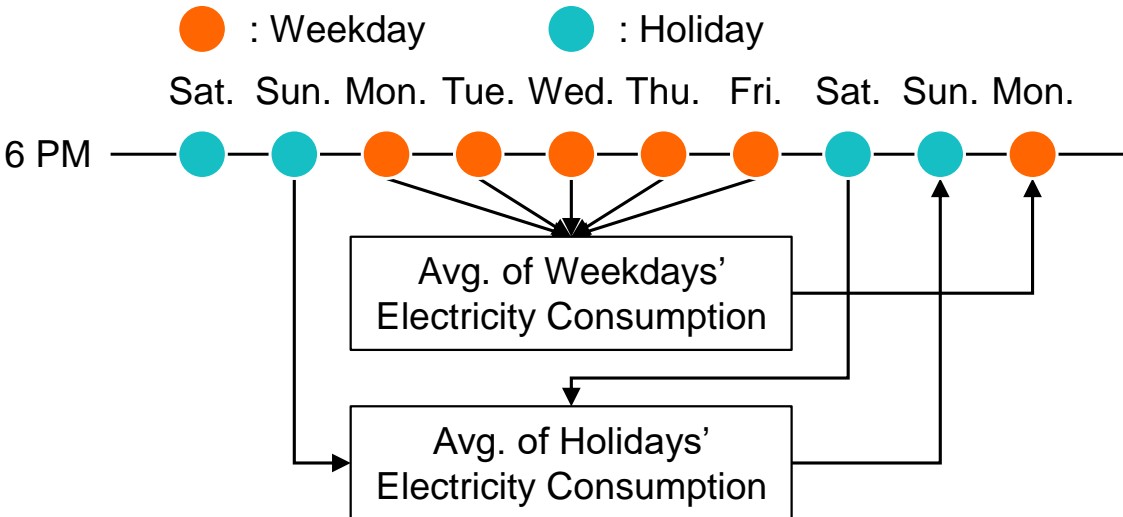

**Figure 5.** Average electricity use per hour for each day of the week and holidays.

**Table 3.** Input variables and their information for BiGTA-net modeling.

| Variable Name | Data Type | Data Format | Description |
|---|---|---|---|
| $Hour_x$ | Numeric | Timestamp information | Sine value of the hour |
| $Hour_y$ | Numeric | Timestamp information | Cosine value of the hour |
| $DOTW_x$ | Numeric | Timestamp information | Sine value of the day of the week |
| $DOTW_y$ | Numeric | Timestamp information | Cosine value of the day of the week |
| Holi | binary | Timestamp information | Holiday indicator for holiday |
| Temp | Numeric | Climate Data | Ambient temperature |
| Humi | Numeric | Climate Data | Relative humidity |
| WS | Numeric | Climate Data | Wind velocity |
| THI | Numeric | Climate Data | Temperature–humidity index |
| WCT | Numeric | Climate Data | Wind chill temperature |
| $Cons_1$ | Numeric | Past Power Consumption | Power consumption one day prior |
| $Holi_1$ | binary | Past Power Consumption | Holiday indicator one day prior |
| $Cons_7$ | Numeric | Past Power Consumption | Power consumption one week prior |
| $Holi_7$ | binary | Past Power Consumption | Holiday indicator one week prior |
| $Cons_{avg}$ | numeric | Past Power Consumption | Average weekly power consumption |

*2.2. BiGTA-Net Modeling*

The BiGTA-net model, illustrated in Figure 6, presents a meticulously crafted hybrid architecture that adeptly merges the advantages of both Bi-GRU and TCN, effectively transcending their respective limitations. The primary objective is to formulate a three-stage prediction model that systematically enhances predictive accuracy by harnessing the inherent strengths of these constituent components. To achieve this objective, a significant attention mechanism is seamlessly integrated to facilitate the harmonious fusion of Bi-GRU and TCN. This orchestrated synergy serves the purpose of constructing a predictive model for building electricity consumption that boasts high accuracy and encompasses multiple stages of prediction refinement. For an in-depth comprehension of the theoretical foundations underpinning Bi-GRU and TCN, readers are referred to Appendix A, which provides comprehensive details. This supplementary resource offers a thorough exploration of the conceptual underpinnings, operational principles, and pertinent prior research pertaining to these two pivotal elements within the model.

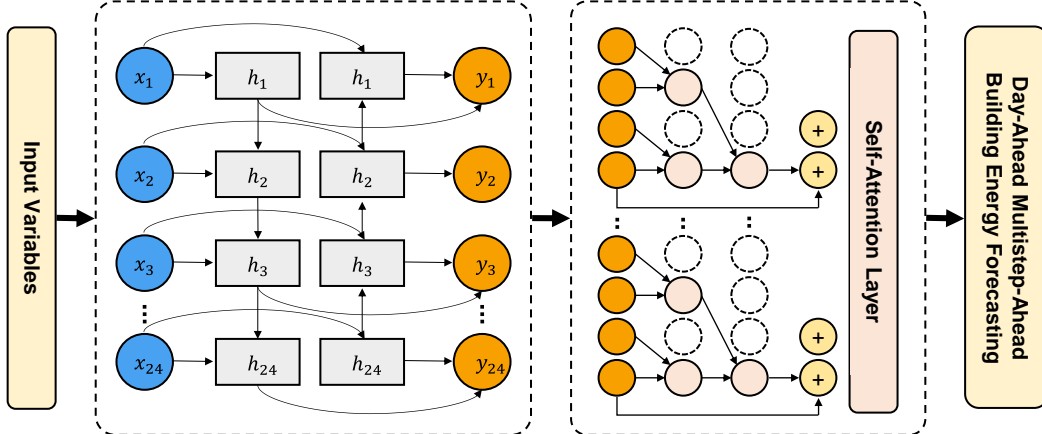

**Figure 6.** System architecture of BiGTA-net.

### 2.2.1. Bidirectional Gated Recurrent Unit

The modeling journey commences with the Bi-GRU, an advancement over traditional RNNs designed to excel in processing sequential time-series data. While conventional RNNs are recognized for their capability to recall historical sequences, they have encountered challenges such as gradient vanishing and exploding. To address these challenges, the GRU was introduced, incorporating specialized gating mechanisms to effectively manage long-term data dependencies [42]. Within the architecture, two distinct GRUs—forward and backward GRUs—are integrated to compose the Bi-GRU, enabling a comprehensive analysis of sequence dynamics [43]. Despite its computationally demanding dual-structured design, this two-pronged approach empowers the model to discern intricate temporal patterns. For an in-depth comprehension of the mathematical intricacies underpinning the Bi-GRU's design, readers are referred to the extensive elaboration in the Keras official documentation [44].

### 2.2.2. Temporal Convolutional Network

The TCN emerges as a groundbreaking solution tailored explicitly for time-series data processing, offering a countermeasure to challenges encountered by sequential models such as the Bi-GRU. TCN employs causal convolutions at its core, ensuring predictions rely solely on current and past data, preserving the temporal sequence's integrity [45]. A defining characteristic of TCNs is their adeptness in capturing long-term patterns through dilated convolutions. These convolutions expand the network's receptive field by introducing fixed steps between neighboring filter taps, enhancing computational efficiency while capturing extended dependencies [46]. The TCN architecture also incorporates residual blocks, addressing the vanishing gradient problem and ensuring stable learning across layers. TCN's adaptability to varying sequence lengths and seamless integration with Bi-GRU outputs form a hierarchical structure that boosts computational efficiency and learning potential. However, TCN's lack of inherent consideration for future data points can impact tasks with significant forward-looking dependencies.

### 2.2.3. Attention Mechanism

The innovation becomes prominent through the introduction of the attention mechanism, a dynamic concept within the realm of deep learning. This mechanism assigns significance or 'attention' to specific segments of sequences, ensuring the model captures essential features for precise predictions. Within the context of the BiGTA-net architecture, this concept has been ingeniously adapted, resulting in a distinctive approach that seamlessly integrates Bi-GRU, TCN, and the attention mechanism. The attention mechanism introduced is referred to as the dual-stage self-attention mechanism (DSSAM), situated at the junction of TCN's output and the subsequent stages of the model [47]. By establishing

correlations across various time steps and dimensions, the DSSAM enhances computational efficiency while strategically highlighting informative features.

The role of the attention mechanism is pivotal in refining the output generated by TCN. Instead of treating all features uniformly, it adeptly identifies and amplifies the most relevant and predictive elements. This dynamic allocation of attention ensures that while the Bi-GRU captures temporal dynamics and the TCN captures long-term dependencies, the attention mechanism focuses on crucial features. As a result, the model achieves enhanced predictive capabilities by synergizing the strengths of Bi-GRU, TCN, and the attention mechanism. The approach incorporates the utilization of the scaled exponential linear units (SELU) [48] activation function, a strategic choice made to address challenges linked to long-term dependencies and gradient vanishing. This integration of SELU enhances stability in the learning process and ultimately contributes to more accurate predictions [49].

## 3. Results and Discussion

### 3.1. Evaluation Criteria

To evaluate the predictive capabilities of the forecasting model, a variety of performance metrics were utilized, including mean absolute percentage error (MAPE), root mean square error (RMSE), and mean absolute error (MAE). These metrics hold widespread recognition and offer a robust assessment of prediction accuracy [50].

The MAPE serves as a valuable statistical measure of prediction accuracy, particularly pertinent in the context of trend forecasting. This metric quantifies the error as a percentage, rendering the outcomes intuitively interpretable. While the MAPE may become inflated when actual values approach zero, this circumstance does not apply to the dataset under consideration. The calculation of MAPE is performed using Equation (7).

$$\mathrm{MAPE} = 1/n \times \sum |(Y_t - \hat{Y}_t)/Y_t| \times 100, \tag{7}$$

where $Y_t$ and $\hat{Y}_t$ represent the actual and predicted values, respectively, and n represents the total number of observations.

The RMSE, or the root mean square deviation, aggregates the residuals to provide a single metric of predictive capability. The RMSE, calculated using Equation (8), is the square root of the average squared differences between the forecast values ($\hat{Y}_t$) and the actual values ($Y_t$). The RMSE equals the standard deviation for an unbiased estimator, indicating the standard error.

$$\mathrm{RMSE} = \sqrt{(1/n \times \sum (Y_t - \hat{Y}_t)^2)}. \tag{8}$$

The MAE is a statistical measure used to gauge the proximity of predictions or forecasts to the eventual outcomes. This metric is calculated by considering the average of the absolute differences between the predicted and actual values. Equation (9) outlines the calculation for the MAE.

$$\mathrm{MAE} = 1/n \times \sum |Y_t - \hat{Y}_t|. \tag{9}$$

### 3.2. Experimental Design

The experiments were conducted in an environment that utilized Python (v.3.8) [51], complemented by machine learning libraries such as scikit-learn (v.1.2.1) [52] and Keras (v.2.9.0) [44,53]. The computational resources included an 11th Gen Intel(R) Core(TM) i9-11900KF CPU operating at 3.50 GHz, an NVIDIA GeForce RTX 3070 GPU, and 64.0GB of RAM. The proposed BiGTA-net model was evaluated against various well-regarded RNN models, such as LSTM, Bi-LSTM, GRU, and GRU-TCN. The hyperparameters were standardized across all models to ensure a fair and balanced comparison. This approach minimized potential bias in the evaluation results due to model-specific preferences or advantageous parameter settings. The common set of hyperparameters for all the models included 25 training epochs, a batch size of 24, and the Adam optimizer with a learning

rate of 0.001 [54]. The MAE was chosen as the key metric for evaluating the performance of the models, providing a standardized measure of comparison.

The training dataset for the BiGTA-net model was constructed by utilizing hourly electrical consumption data from 1 to 7 March 2015, for the educational building dataset, and from 11 to 17 January 2016, for the AEP dataset. In the case of the educational building dataset, the data spanning from 8 March 2015, to 28 February 2019, was allocated for training, while the subsequent period, 1 March 2019, to 29 February 2020, was designated as the testing set. For the AEP dataset, data ranging from 18 January to 30 April 2016, was employed for training purposes, with the timeframe between 1 and 27 May 2016, reserved for testing. The dataset was partitioned into training (in-sample) and testing (out-of-sample) subsets, maintaining an 80:20 ratio. Prior to the division, min–max scaling was applied to the training data, standardizing the raw electricity consumption values within a specific range. This scaling transformation was subsequently extended to the testing data, ensuring uniformity in the range of both training and testing datasets. This process ensured that the original data scale did not influence the model's performance.

### 3.3. Experimental Results

In the experimental outcomes, the performances of diverse model configurations were initially investigated, as presented in Table 4. Specifically, a total of 16 models with varying network architectures, activation functions, and the incorporation of the attention mechanism were evaluated. Among the specifics detailed in Table 4, the prominent focus is on the Bi-GRU-TCN-I model, alternatively known as BiGTA-net, which was proposed in this study. This particular model embraced the Bi-GRU-TCN architecture, utilized the SELU activation function, and integrated the attention mechanism, setting it apart from the remaining models.

**Table 4.** Comparison of hybrid deep learning model architectures.

| Models | Neural Network | Activation Function | Attention Mechanism |
|---|---|---|---|
| LSTM-TCN-I<br>LSTM-TCN-II | LSTM-TCN | SELU | O<br>X |
| LSTM-TCN-III<br>LSTM-TCN-IV | | ReLU | O<br>X |
| Bi-LSTM-TCN-I<br>Bi-LSTM-TCN-II | Bi-LSTM-TCN | SELU | O<br>X |
| Bi-LSTM-TCN-III<br>Bi-LSTM-TCN-IV | | ReLU | O<br>X |
| GRU-TCN-I<br>GRU-TCN-II | GRU-TCN | SELU | O<br>X |
| GRU-TCN-III<br>GRU-TCN-IV | | ReLU | O<br>X |
| Bi-GRU-TCN-I<br>Bi-GRU-TCN-II | Bi-GRU-TCN | SELU | O<br>X |
| Bi-GRU-TCN-III<br>Bi-GRU-TCN-IV | | ReLU | O<br>X |

The performance of these models was evaluated using three key metrics: MAPE, RMSE, and MAE, as presented in Tables 5–10. The experimental results were divided into two main categories, results obtained from the educational building dataset and the AEP dataset.

**Table 5.** MAPE comparison for the educational building dataset.

| Step | LSTM-TCN | | | | Bi-LSTM-TCN | | | | GRU-TCN | | | | Bi-GRU-TCN | | | |
|---|---|---|---|---|---|---|---|---|---|---|---|---|---|---|---|---|
| | I | II | III | IV | I | II | III | IV | I | II | III | IV | I | II | III | IV |
| 1 | 4.51 | 5.14 | 4.09 | 23.36 | 5.61 | 7.38 | 9.83 | 22.86 | 4.64 | 4.66 | 4.35 | 5.04 | 3.88 | 3.59 | 4.64 | 4.66 |
| 2 | 4.93 | 5.64 | 4.74 | 23.35 | 5.83 | 7.04 | 10.33 | 23.03 | 4.98 | 4.70 | 5.18 | 5.00 | 4.49 | 4.41 | 4.98 | 4.70 |
| 3 | 5.00 | 5.78 | 4.80 | 23.39 | 6.21 | 7.37 | 8.45 | 23.04 | 5.37 | 4.92 | 5.57 | 5.36 | 4.76 | 4.39 | 5.37 | 4.92 |
| 4 | 5.37 | 6.20 | 5.14 | 23.40 | 6.38 | 7.30 | 7.96 | 15.38 | 5.61 | 5.16 | 5.75 | 5.50 | 4.94 | 4.68 | 5.61 | 5.16 |
| 5 | 5.76 | 6.40 | 5.34 | 23.33 | 6.29 | 7.33 | 7.40 | 12.16 | 5.76 | 5.70 | 5.74 | 5.92 | 5.12 | 5.09 | 5.76 | 5.70 |
| 6 | 5.89 | 6.69 | 5.45 | 23.17 | 6.29 | 7.71 | 6.87 | 11.20 | 5.92 | 5.75 | 5.94 | 5.87 | 5.25 | 5.21 | 5.92 | 5.75 |
| 7 | 5.89 | 6.78 | 5.68 | 22.94 | 6.28 | 7.92 | 6.99 | 11.65 | 6.01 | 6.09 | 5.82 | 6.09 | 5.32 | 5.18 | 6.01 | 6.09 |
| 8 | 5.81 | 7.01 | 5.61 | 22.66 | 6.62 | 8.62 | 7.04 | 13.19 | 6.11 | 6.04 | 5.97 | 6.15 | 5.42 | 5.19 | 6.11 | 6.04 |
| 9 | 5.87 | 7.13 | 5.61 | 22.38 | 6.43 | 9.08 | 7.54 | 13.61 | 6.23 | 6.49 | 6.07 | 6.38 | 5.48 | 5.38 | 6.23 | 6.49 |
| 10 | 5.76 | 7.47 | 5.50 | 22.11 | 6.51 | 8.63 | 7.49 | 14.10 | 6.02 | 6.70 | 6.00 | 6.43 | 5.52 | 5.35 | 6.02 | 6.70 |
| 11 | 5.89 | 7.77 | 5.61 | 21.95 | 6.41 | 8.44 | 7.49 | 14.04 | 6.09 | 6.84 | 6.15 | 6.53 | 5.53 | 5.42 | 6.09 | 6.84 |
| 12 | 6.00 | 7.83 | 5.61 | 21.88 | 6.49 | 7.74 | 7.88 | 14.06 | 6.20 | 7.06 | 6.06 | 6.72 | 5.52 | 5.46 | 6.20 | 7.06 |
| 13 | 6.14 | 8.11 | 5.74 | 21.88 | 6.47 | 7.53 | 8.04 | 12.66 | 6.42 | 7.03 | 6.09 | 6.60 | 5.66 | 5.57 | 6.42 | 7.03 |
| 14 | 6.14 | 8.15 | 5.81 | 21.94 | 6.51 | 7.73 | 9.19 | 13.16 | 6.42 | 7.40 | 6.15 | 6.79 | 5.65 | 5.52 | 6.42 | 7.40 |
| 15 | 6.34 | 8.16 | 5.92 | 22.02 | 6.73 | 7.83 | 8.53 | 12.25 | 6.54 | 7.27 | 6.19 | 7.08 | 5.81 | 5.47 | 6.54 | 7.27 |
| 16 | 6.23 | 8.18 | 5.97 | 22.10 | 7.03 | 8.67 | 8.65 | 12.58 | 6.61 | 7.41 | 6.12 | 7.15 | 5.66 | 5.44 | 6.61 | 7.41 |
| 17 | 6.44 | 8.02 | 6.01 | 22.17 | 7.17 | 8.53 | 8.05 | 13.37 | 6.55 | 7.59 | 6.32 | 7.23 | 5.66 | 5.74 | 6.55 | 7.59 |
| 18 | 6.39 | 7.93 | 6.24 | 22.28 | 7.34 | 9.18 | 7.82 | 13.92 | 6.48 | 7.61 | 6.35 | 7.33 | 5.49 | 5.70 | 6.48 | 7.61 |
| 19 | 6.46 | 7.91 | 6.00 | 22.50 | 7.74 | 9.76 | 7.91 | 15.28 | 6.46 | 7.88 | 6.31 | 7.24 | 5.59 | 5.70 | 6.46 | 7.88 |
| 20 | 6.44 | 7.85 | 6.03 | 22.77 | 7.90 | 10.08 | 8.01 | 17.06 | 6.72 | 7.41 | 6.43 | 7.43 | 5.56 | 5.76 | 6.72 | 7.41 |
| 21 | 6.65 | 7.84 | 5.93 | 23.05 | 8.25 | 9.56 | 8.01 | 18.93 | 6.61 | 7.20 | 6.32 | 7.38 | 5.59 | 6.01 | 6.61 | 7.20 |
| 22 | 6.49 | 7.91 | 5.99 | 23.30 | 8.49 | 10.23 | 7.78 | 21.15 | 6.63 | 7.09 | 6.29 | 7.11 | 5.67 | 6.18 | 6.63 | 7.09 |
| 23 | 6.32 | 7.81 | 5.76 | 23.50 | 8.31 | 10.20 | 8.13 | 23.95 | 6.68 | 7.03 | 6.28 | 6.86 | 5.66 | 6.39 | 6.68 | 7.03 |
| 24 | 5.97 | 7.61 | 5.60 | 23.62 | 8.44 | 10.88 | 8.66 | 25.87 | 6.57 | 6.81 | 6.22 | 6.86 | 5.65 | 6.45 | 6.57 | 6.81 |
| Avg. | 5.95 | 7.31 | 5.59 | 22.71 | 6.90 | 8.53 | 8.09 | 16.19 | 6.15 | 6.58 | 5.99 | 6.50 | 5.37 | 5.39 | 6.15 | 6.58 |

**Table 6.** RMSE comparison for the educational building dataset.

| Step | LSTM-TCN | | | | Bi-LSTM-TCN | | | | GRU-TCN | | | | Bi-GRU-TCN | | | |
|---|---|---|---|---|---|---|---|---|---|---|---|---|---|---|---|---|
| | I | II | III | IV | I | II | III | IV | I | II | III | IV | I | II | III | IV |
| 1 | 134.8 | 144.7 | 128.8 | 704.6 | 165.4 | 233.2 | 307.3 | 644.7 | 140.1 | 141.8 | 133.9 | 156.2 | 118.8 | 110.7 | 140.1 | 141.8 |
| 2 | 148.8 | 164.6 | 152.7 | 699.9 | 172.4 | 218.9 | 296.3 | 621.0 | 150.0 | 149.9 | 155.7 | 162.1 | 140.4 | 136.7 | 150.0 | 149.9 |
| 3 | 154.2 | 170.5 | 156.8 | 696.8 | 181.0 | 223.8 | 254.9 | 778.4 | 159.6 | 156.9 | 166.4 | 167.2 | 151.1 | 141.5 | 159.6 | 156.9 |
| 4 | 164.5 | 179.1 | 163.9 | 694.2 | 185.2 | 216.6 | 233.0 | 439.3 | 165.2 | 162.1 | 171.7 | 173.0 | 158.6 | 149.5 | 165.2 | 162.1 |
| 5 | 175.6 | 188.6 | 169.2 | 691.4 | 185.1 | 224.3 | 217.7 | 331.3 | 170.5 | 176.1 | 175.2 | 181.7 | 164.5 | 160.7 | 170.5 | 176.1 |
| 6 | 181.4 | 195.6 | 177.3 | 688.4 | 185.4 | 230.9 | 219.1 | 317.3 | 176.7 | 180.4 | 181.3 | 183.1 | 169.1 | 164.3 | 176.7 | 180.4 |
| 7 | 181.5 | 200.4 | 182.6 | 685.2 | 187.8 | 234.8 | 210.7 | 349.7 | 180.2 | 188.5 | 179.7 | 188.4 | 174.4 | 164.3 | 180.2 | 188.5 |
| 8 | 179.1 | 206.5 | 182.3 | 682.3 | 198.6 | 245.9 | 216.4 | 397.0 | 184.1 | 189.8 | 182.4 | 192.9 | 176.6 | 165.3 | 184.1 | 189.8 |
| 9 | 179.2 | 215.0 | 184.9 | 680.0 | 198.9 | 251.7 | 232.1 | 399.7 | 184.5 | 200.1 | 184.3 | 200.7 | 175.8 | 168.2 | 184.5 | 200.1 |
| 10 | 175.9 | 223.4 | 179.6 | 678.2 | 203.3 | 240.5 | 236.1 | 397.2 | 180.6 | 207.7 | 182.2 | 201.9 | 173.7 | 168.3 | 180.6 | 207.7 |
| 11 | 176.9 | 232.4 | 183.1 | 677.3 | 205.6 | 235.8 | 230.3 | 411.5 | 180.7 | 213.1 | 185.2 | 204.7 | 173.1 | 170.6 | 180.7 | 213.1 |
| 12 | 179.5 | 238.3 | 182.2 | 677.1 | 209.9 | 222.7 | 235.8 | 409.0 | 182.4 | 217.5 | 184.0 | 208.1 | 173.7 | 170.9 | 182.4 | 217.5 |
| 13 | 181.6 | 243.3 | 184.9 | 677.5 | 213.0 | 221.4 | 235.6 | 363.7 | 186.0 | 219.3 | 185.7 | 204.1 | 175.7 | 173.2 | 186.0 | 219.3 |
| 14 | 183.1 | 245.9 | 183.6 | 678.6 | 212.8 | 232.1 | 262.7 | 363.3 | 186.9 | 226.1 | 186.5 | 207.6 | 178.3 | 172.8 | 186.9 | 226.1 |
| 15 | 187.0 | 242.6 | 187.2 | 680.9 | 219.3 | 239.2 | 242.0 | 362.5 | 188.2 | 222.6 | 187.3 | 214.4 | 183.1 | 173.3 | 188.2 | 222.6 |
| 16 | 185.0 | 239.9 | 191.3 | 684.1 | 226.0 | 262.9 | 254.7 | 418.2 | 191.3 | 223.3 | 186.6 | 217.3 | 180.6 | 173.6 | 191.3 | 223.3 |
| 17 | 188.2 | 233.6 | 182.8 | 688.0 | 232.5 | 258.0 | 237.7 | 485.0 | 188.3 | 228.8 | 188.3 | 217.8 | 181.3 | 177.2 | 188.3 | 228.8 |
| 18 | 188.6 | 230.7 | 190.4 | 693.0 | 232.7 | 278.9 | 228.0 | 532.6 | 188.5 | 227.4 | 191.4 | 219.3 | 178.6 | 178.3 | 188.5 | 227.4 |
| 19 | 189.9 | 228.4 | 184.4 | 699.2 | 238.8 | 287.7 | 237.1 | 608.7 | 190.5 | 233.0 | 192.1 | 218.4 | 180.5 | 181.4 | 190.5 | 233.0 |
| 20 | 189.9 | 225.4 | 183.1 | 706.2 | 240.3 | 288.0 | 231.4 | 699.6 | 195.3 | 223.2 | 197.3 | 223.4 | 180.3 | 183.0 | 195.3 | 223.2 |
| 21 | 194.8 | 223.1 | 182.4 | 713.6 | 247.4 | 270.8 | 243.1 | 797.5 | 194.8 | 221.1 | 192.5 | 219.3 | 180.2 | 188.8 | 194.8 | 221.1 |

**Table 6.** *Cont.*

| Step | LSTM-TCN | | | | Bi-LSTM-TCN | | | | GRU-TCN | | | | Bi-GRU-TCN | | | |
|------|------|------|------|------|------|------|------|------|------|------|------|------|------|------|------|------|
| | I | II | III | IV | I | II | III | IV | I | II | III | IV | I | II | III | IV |
| 22 | 193.6 | 224.2 | 184.4 | 720.0 | 251.0 | 290.3 | 244.4 | 879.6 | 195.8 | 217.7 | 193.7 | 211.8 | 182.6 | 192.7 | 195.8 | 217.7 |
| 23 | 193.6 | 221.2 | 181.0 | 725.3 | 244.2 | 285.5 | 264.6 | 971.5 | 197.6 | 213.7 | 189.8 | 206.3 | 180.6 | 199.1 | 197.6 | 213.7 |
| 24 | 192.9 | 221.0 | 177.5 | 729.6 | 246.6 | 316.9 | 294.4 | 863.1 | 198.7 | 212.5 | 189.8 | 208.1 | 180.4 | 203.3 | 198.7 | 212.5 |
| Avg. | 179.1 | 214.1 | 177.4 | 693.8 | 211.8 | 250.5 | 244.4 | 535.1 | 181.5 | 202.2 | 181.8 | 199.5 | 171.3 | 169.5 | 181.5 | 202.2 |

**Table 7.** MAE comparison for the educational building dataset.

| Step | LSTM-TCN | | | | Bi-LSTM-TCN | | | | GRU-TCN | | | | Bi-GRU-TCN | | | |
|------|------|------|------|------|------|------|------|------|------|------|------|------|------|------|------|------|
| | I | II | III | IV | I | II | III | IV | I | II | III | IV | I | II | III | IV |
| 1 | 99.2 | 113.1 | 93.7 | 559.0 | 128.2 | 168.2 | 214.7 | 527.0 | 103.5 | 107.8 | 97.5 | 118.2 | 86.4 | 80.3 | 103.5 | 107.8 |
| 2 | 109.3 | 126.8 | 110.5 | 557.4 | 132.0 | 158.2 | 222.0 | 501.2 | 110.9 | 110.0 | 115.9 | 119.3 | 100.5 | 99.4 | 110.9 | 110.0 |
| 3 | 112.5 | 130.0 | 112.0 | 557.3 | 138.6 | 166.2 | 188.8 | 578.3 | 119.5 | 114.4 | 124.4 | 124.1 | 107.2 | 99.9 | 119.5 | 114.4 |
| 4 | 120.8 | 137.9 | 118.1 | 557.1 | 142.0 | 163.3 | 175.6 | 343.8 | 123.8 | 118.6 | 128.2 | 127.5 | 111.7 | 105.9 | 123.8 | 118.6 |
| 5 | 129.3 | 144.7 | 122.2 | 555.5 | 140.8 | 167.5 | 162.4 | 259.4 | 127.0 | 130.5 | 129.1 | 136.0 | 116.5 | 115.3 | 127.0 | 130.5 |
| 6 | 133.0 | 150.6 | 126.5 | 552.3 | 139.9 | 175.2 | 157.4 | 245.5 | 131.4 | 133.0 | 134.1 | 135.1 | 119.8 | 118.1 | 131.4 | 133.0 |
| 7 | 132.9 | 153.8 | 131.9 | 548.0 | 141.4 | 179.3 | 156.1 | 263.4 | 134.0 | 140.6 | 131.8 | 139.7 | 122.1 | 117.5 | 134.0 | 140.6 |
| 8 | 130.9 | 159.1 | 131.1 | 543.0 | 150.8 | 192.4 | 160.3 | 300.4 | 136.6 | 140.5 | 134.2 | 142.4 | 123.8 | 117.3 | 136.6 | 140.5 |
| 9 | 130.9 | 164.9 | 132.1 | 538.0 | 149.1 | 199.5 | 170.4 | 305.9 | 137.7 | 150.7 | 136.1 | 149.3 | 124.4 | 120.7 | 137.7 | 150.7 |
| 10 | 128.0 | 172.7 | 127.6 | 533.5 | 152.3 | 189.9 | 172.8 | 313.0 | 133.4 | 157.2 | 133.8 | 150.3 | 123.9 | 120.4 | 133.4 | 157.2 |
| 11 | 130.2 | 180.6 | 130.5 | 531.1 | 152.3 | 185.6 | 169.7 | 320.3 | 134.2 | 161.4 | 137.1 | 152.5 | 123.6 | 122.7 | 134.2 | 161.4 |
| 12 | 132.9 | 183.7 | 130.1 | 530.7 | 155.0 | 171.7 | 177.2 | 318.7 | 136.3 | 166.1 | 135.7 | 156.6 | 123.8 | 123.1 | 136.3 | 166.1 |
| 13 | 135.2 | 190.0 | 133.2 | 531.3 | 155.7 | 169.0 | 179.4 | 282.4 | 140.9 | 166.5 | 137.2 | 153.2 | 126.4 | 125.7 | 140.9 | 166.5 |
| 14 | 135.6 | 191.1 | 133.1 | 532.7 | 155.9 | 175.6 | 202.5 | 286.3 | 141.0 | 174.2 | 137.7 | 157.3 | 127.1 | 124.9 | 141.0 | 174.2 |
| 15 | 140.1 | 189.6 | 136.1 | 534.4 | 161.4 | 178.8 | 187.7 | 270.3 | 142.9 | 171.0 | 138.5 | 164.0 | 131.5 | 124.3 | 142.9 | 171.0 |
| 16 | 137.5 | 187.9 | 137.9 | 535.8 | 168.7 | 200.4 | 194.6 | 288.6 | 145.3 | 172.9 | 136.9 | 166.4 | 128.9 | 123.9 | 145.3 | 172.9 |
| 17 | 141.0 | 183.6 | 134.6 | 537.0 | 174.0 | 196.8 | 178.5 | 318.2 | 143.2 | 177.6 | 140.6 | 167.8 | 129.5 | 129.4 | 143.2 | 177.6 |
| 18 | 140.3 | 181.2 | 138.7 | 539.2 | 176.5 | 214.8 | 173.3 | 337.9 | 143.1 | 177.1 | 142.3 | 169.6 | 126.6 | 129.6 | 143.1 | 177.1 |
| 19 | 141.7 | 179.9 | 134.5 | 543.8 | 184.5 | 227.3 | 174.5 | 382.7 | 144.0 | 183.6 | 141.6 | 168.3 | 128.6 | 131.6 | 144.0 | 183.6 |
| 20 | 141.4 | 177.7 | 133.5 | 549.5 | 187.4 | 231.5 | 176.1 | 442.3 | 149.3 | 173.1 | 145.9 | 172.8 | 128.3 | 133.1 | 149.3 | 173.1 |
| 21 | 145.6 | 176.4 | 132.8 | 555.6 | 195.0 | 215.2 | 177.9 | 507.0 | 148.1 | 169.8 | 141.8 | 169.9 | 128.5 | 139.2 | 148.1 | 169.8 |
| 22 | 143.2 | 177.4 | 133.9 | 561.4 | 199.4 | 230.0 | 175.4 | 574.6 | 148.5 | 166.9 | 142.6 | 163.4 | 130.1 | 143.0 | 148.5 | 166.9 |
| 23 | 141.6 | 174.8 | 129.6 | 565.8 | 193.5 | 227.1 | 184.3 | 652.7 | 149.8 | 164.1 | 139.7 | 157.3 | 129.3 | 148.6 | 149.8 | 164.1 |
| 24 | 136.2 | 172.4 | 125.8 | 568.5 | 194.9 | 247.7 | 199.4 | 659.3 | 148.8 | 160.2 | 137.9 | 157.7 | 128.7 | 150.9 | 148.8 | 160.2 |
| Avg. | 132.0 | 166.7 | 127.9 | 546.6 | 161.2 | 193.0 | 180.5 | 386.6 | 136.4 | 153.7 | 134.2 | 150.8 | 122.0 | 122.7 | 136.4 | 153.7 |

**Table 8.** MAPE comparison for the AEP dataset.

| Step | LSTM-TCN | | | | Bi-LSTM-TCN | | | | GRU-TCN | | | | Bi-GRU-TCN | | | |
|------|------|------|------|------|------|------|------|------|------|------|------|------|------|------|------|------|
| | I | II | III | IV | I | II | III | IV | I | II | III | IV | I | II | III | IV |
| 1 | 29.34 | 28.39 | 28.18 | 33.29 | 33.05 | 35.48 | 36.61 | 55.79 | 26.86 | 29.18 | 30.12 | 27.24 | 23.78 | 24.64 | 26.86 | 29.18 |
| 2 | 28.85 | 26.18 | 27.53 | 32.84 | 31.20 | 32.03 | 34.83 | 37.99 | 26.06 | 28.21 | 29.93 | 28.85 | 25.62 | 26.84 | 26.06 | 28.21 |
| 3 | 27.83 | 25.92 | 27.82 | 29.84 | 32.38 | 30.19 | 36.26 | 77.86 | 26.32 | 30.59 | 29.68 | 28.22 | 26.45 | 27.15 | 26.32 | 30.59 |
| 4 | 26.99 | 27.04 | 28.15 | 28.23 | 30.96 | 30.28 | 36.39 | 50.02 | 27.62 | 29.40 | 29.35 | 29.01 | 27.23 | 30.36 | 27.62 | 29.40 |
| 5 | 26.47 | 31.27 | 29.39 | 28.02 | 28.14 | 31.88 | 34.96 | 32.90 | 29.41 | 28.42 | 29.08 | 29.12 | 26.52 | 28.81 | 29.41 | 28.42 |
| 6 | 26.22 | 30.77 | 29.10 | 27.67 | 28.04 | 27.54 | 35.89 | 77.49 | 30.12 | 29.38 | 28.68 | 30.40 | 26.22 | 29.80 | 30.12 | 29.38 |
| 7 | 26.16 | 30.52 | 29.62 | 27.34 | 26.05 | 35.42 | 32.58 | 44.10 | 30.29 | 29.83 | 28.36 | 30.58 | 26.79 | 30.09 | 30.29 | 29.83 |
| 8 | 26.29 | 35.94 | 28.40 | 26.71 | 25.73 | 30.26 | 35.78 | 32.66 | 30.61 | 29.99 | 27.99 | 30.58 | 28.87 | 30.24 | 30.61 | 29.99 |
| 9 | 26.53 | 32.77 | 29.12 | 27.27 | 25.60 | 27.77 | 34.22 | 76.03 | 29.29 | 28.63 | 27.86 | 29.53 | 27.06 | 29.96 | 29.29 | 28.63 |
| 10 | 27.32 | 29.69 | 27.48 | 27.54 | 26.61 | 27.36 | 35.70 | 36.91 | 29.02 | 27.72 | 27.90 | 28.69 | 27.18 | 29.22 | 29.02 | 27.72 |
| 11 | 27.51 | 29.95 | 28.11 | 27.44 | 26.82 | 29.70 | 32.24 | 38.45 | 29.10 | 26.61 | 28.08 | 27.97 | 27.14 | 29.48 | 29.10 | 26.61 |

**Table 8.** *Cont.*

| Step | LSTM-TCN | | | | Bi-LSTM-TCN | | | | GRU-TCN | | | | Bi-GRU-TCN | | | |
|------|------|------|------|------|------|------|------|------|------|------|------|------|------|------|------|------|
| | I | II | III | IV | I | II | III | IV | I | II | III | IV | I | II | III | IV |
| 12 | 27.79 | 28.37 | 27.81 | 26.30 | 27.85 | 32.31 | 31.44 | 72.68 | 28.10 | 26.23 | 28.28 | 27.72 | 26.95 | 30.28 | 28.10 | 26.23 |
| 13 | 28.13 | 27.94 | 27.97 | 27.06 | 35.03 | 33.45 | 33.63 | 44.07 | 27.29 | 25.37 | 28.63 | 26.34 | 26.04 | 29.86 | 27.29 | 25.37 |
| 14 | 28.40 | 28.50 | 28.67 | 26.54 | 27.46 | 32.83 | 31.19 | 33.48 | 27.05 | 25.30 | 29.03 | 26.88 | 26.40 | 28.80 | 27.05 | 25.30 |
| 15 | 29.26 | 27.95 | 29.31 | 28.29 | 30.34 | 32.13 | 31.96 | 68.84 | 28.02 | 24.65 | 29.47 | 26.73 | 27.23 | 30.16 | 28.02 | 24.65 |
| 16 | 29.73 | 28.34 | 29.15 | 31.27 | 27.45 | 31.19 | 32.80 | 34.33 | 28.07 | 24.99 | 29.74 | 26.69 | 27.10 | 29.97 | 28.07 | 24.99 |
| 17 | 29.78 | 28.81 | 29.09 | 28.16 | 27.11 | 30.01 | 34.39 | 30.08 | 28.88 | 24.59 | 30.15 | 25.71 | 26.48 | 28.68 | 28.88 | 24.59 |
| 18 | 30.03 | 28.72 | 28.18 | 26.33 | 28.20 | 30.05 | 31.65 | 59.95 | 28.63 | 24.89 | 30.44 | 25.67 | 26.35 | 27.93 | 28.63 | 24.89 |
| 19 | 29.51 | 29.10 | 28.17 | 25.18 | 28.26 | 30.33 | 33.12 | 30.48 | 28.58 | 25.47 | 30.62 | 26.51 | 26.37 | 28.31 | 28.58 | 25.47 |
| 20 | 28.37 | 28.61 | 27.54 | 25.31 | 28.64 | 29.65 | 32.98 | 27.73 | 28.40 | 25.75 | 30.64 | 26.40 | 26.71 | 26.29 | 28.40 | 25.75 |
| 21 | 28.14 | 29.26 | 27.30 | 26.92 | 31.00 | 28.73 | 34.29 | 27.75 | 28.46 | 26.94 | 30.73 | 27.51 | 27.26 | 27.53 | 28.46 | 26.94 |
| 22 | 29.03 | 29.48 | 27.38 | 27.93 | 30.83 | 28.07 | 33.48 | 30.32 | 28.58 | 27.27 | 30.53 | 27.96 | 27.64 | 27.03 | 28.58 | 27.27 |
| 23 | 30.83 | 29.50 | 27.48 | 31.79 | 31.38 | 28.27 | 31.75 | 35.26 | 28.84 | 28.32 | 30.37 | 29.33 | 27.81 | 28.06 | 28.84 | 28.32 |
| 24 | 33.64 | 30.42 | 27.94 | 36.82 | 30.64 | 29.78 | 39.80 | 53.76 | 27.54 | 27.39 | 30.28 | 29.80 | 27.35 | 28.45 | 27.54 | 27.39 |
| Avg. | 28.42 | 29.31 | 28.29 | 28.50 | 29.12 | 30.61 | 34.08 | 46.21 | 28.38 | 27.30 | 29.41 | 28.06 | 26.77 | 28.66 | 28.38 | 27.30 |

**Table 9.** RMSE comparison for the AEP dataset.

| Step | LSTM-TCN | | | | Bi-LSTM-TCN | | | | GRU-TCN | | | | Bi-GRU-TCN | | | |
|------|------|------|------|------|------|------|------|------|------|------|------|------|------|------|------|------|
| | I | II | III | IV | I | II | III | IV | I | II | III | IV | I | II | III | IV |
| 1 | 473.6 | 417.6 | 392.5 | 401.8 | 431.1 | 427.9 | 430.3 | 419.2 | 368.3 | 386.1 | 448.4 | 380.8 | 372.2 | 375.9 | 368.3 | 386.1 |
| 2 | 451.5 | 420.0 | 395.2 | 406.0 | 424.1 | 436.4 | 429.9 | 430.5 | 381.8 | 392.7 | 450.0 | 378.8 | 378.5 | 369.6 | 381.8 | 392.7 |
| 3 | 449.4 | 431.3 | 425.5 | 407.3 | 426.1 | 440.9 | 435.3 | 622.2 | 381.9 | 391.8 | 450.9 | 379.0 | 381.8 | 372.8 | 381.9 | 391.8 |
| 4 | 447.3 | 422.0 | 420.2 | 409.6 | 411.1 | 440.0 | 423.6 | 431.3 | 379.6 | 387.9 | 450.3 | 377.6 | 380.0 | 370.8 | 379.6 | 387.9 |
| 5 | 446.3 | 421.8 | 420.2 | 407.0 | 412.4 | 430.8 | 432.2 | 464.4 | 378.8 | 384.8 | 450.1 | 388.1 | 384.6 | 373.2 | 378.8 | 384.8 |
| 6 | 446.3 | 429.0 | 414.4 | 414.9 | 417.1 | 452.0 | 434.1 | 622.0 | 383.3 | 383.3 | 449.6 | 384.0 | 391.1 | 373.7 | 383.3 | 383.3 |
| 7 | 447.2 | 428.3 | 411.0 | 416.9 | 422.8 | 417.4 | 436.7 | 439.0 | 382.7 | 384.5 | 449.3 | 380.5 | 378.4 | 369.1 | 382.7 | 384.5 |
| 8 | 446.8 | 430.3 | 417.9 | 412.0 | 422.8 | 428.3 | 438.4 | 469.2 | 381.1 | 381.4 | 447.5 | 384.1 | 379.7 | 368.9 | 381.1 | 381.4 |
| 9 | 447.1 | 427.3 | 417.2 | 414.3 | 429.4 | 437.5 | 448.7 | 614.7 | 377.1 | 384.0 | 447.7 | 387.0 | 382.6 | 371.7 | 377.1 | 384.0 |
| 10 | 445.2 | 436.9 | 413.3 | 413.2 | 441.3 | 435.7 | 440.2 | 442.0 | 376.3 | 384.0 | 447.9 | 387.8 | 385.0 | 369.3 | 376.3 | 384.0 |
| 11 | 446.4 | 435.4 | 404.8 | 416.7 | 425.8 | 425.1 | 452.0 | 432.7 | 374.2 | 384.5 | 448.6 | 387.3 | 387.5 | 373.6 | 374.2 | 384.5 |
| 12 | 447.8 | 439.8 | 412.1 | 418.3 | 455.5 | 425.8 | 450.1 | 601.6 | 380.1 | 390.3 | 449.5 | 387.3 | 387.5 | 374.2 | 380.1 | 390.3 |
| 13 | 444.5 | 438.7 | 406.4 | 423.2 | 421.2 | 428.8 | 453.1 | 416.0 | 389.9 | 395.5 | 450.6 | 387.8 | 390.8 | 373.6 | 389.9 | 395.5 |
| 14 | 443.1 | 435.4 | 398.1 | 428.6 | 445.5 | 431.8 | 447.2 | 428.6 | 391.0 | 394.6 | 451.8 | 387.8 | 390.0 | 375.1 | 391.0 | 394.6 |
| 15 | 434.3 | 442.7 | 393.7 | 433.0 | 411.6 | 433.9 | 457.0 | 587.4 | 390.5 | 398.0 | 453.4 | 389.5 | 383.8 | 375.5 | 390.5 | 398.0 |
| 16 | 428.3 | 441.2 | 391.2 | 432.4 | 408.7 | 433.9 | 445.9 | 417.9 | 393.8 | 396.2 | 454.6 | 392.7 | 384.5 | 377.9 | 393.8 | 396.2 |
| 17 | 423.2 | 441.5 | 410.0 | 437.1 | 408.0 | 437.1 | 447.6 | 430.1 | 398.7 | 392.7 | 457.7 | 387.5 | 383.7 | 374.9 | 398.7 | 392.7 |
| 18 | 412.3 | 446.1 | 416.4 | 433.6 | 407.5 | 436.7 | 455.3 | 560.6 | 397.0 | 397.3 | 459.8 | 391.9 | 387.7 | 380.4 | 397.0 | 397.3 |
| 19 | 406.7 | 448.1 | 423.0 | 437.1 | 415.8 | 436.5 | 453.5 | 427.7 | 396.6 | 399.0 | 463.2 | 396.6 | 391.0 | 382.9 | 396.6 | 399.0 |
| 20 | 405.0 | 456.2 | 423.5 | 433.7 | 419.6 | 435.8 | 453.2 | 441.7 | 393.4 | 404.9 | 464.3 | 398.8 | 390.2 | 385.2 | 393.4 | 404.9 |
| 21 | 409.8 | 456.4 | 410.1 | 430.8 | 429.3 | 436.6 | 444.7 | 441.2 | 392.5 | 402.3 | 465.2 | 397.5 | 394.4 | 382.9 | 392.5 | 402.3 |
| 22 | 416.0 | 457.4 | 414.7 | 437.1 | 440.7 | 434.8 | 452.2 | 427.2 | 394.0 | 401.3 | 464.7 | 396.4 | 395.3 | 386.1 | 394.0 | 401.3 |
| 23 | 423.7 | 459.3 | 412.6 | 441.1 | 450.1 | 434.3 | 444.5 | 417.8 | 397.8 | 401.2 | 463.4 | 396.1 | 396.2 | 390.0 | 397.8 | 401.2 |
| 24 | 434.2 | 459.1 | 418.9 | 443.2 | 452.6 | 440.9 | 451.3 | 418.8 | 401.8 | 403.5 | 462.8 | 403.8 | 398.5 | 400.3 | 401.8 | 403.5 |
| Avg. | 436.5 | 438.4 | 410.9 | 422.9 | 426.3 | 434.1 | 444.0 | 475.2 | 386.8 | 392.6 | 454.2 | 388.7 | 386.5 | 377.0 | 386.8 | 392.6 |

In the context of the educational building dataset, the proposed model (Bi-GRU-TCN-I) consistently showcased superior performance in comparison to alternative model configurations. As illustrated in Table 5, the proposed model achieved the lowest MAPE, underscoring its heightened predictive accuracy. Strong corroboration for its superior performance is also substantiated by the findings presented in Tables 6 and 7, where the proposed model demonstrates the least RMSE and MAE values, respectively, signifying a close alignment between the model's predictions and actual values.

**Table 10.** MAE comparison for the AEP dataset.

| Step | LSTM-TCN | | | | Bi-LSTM-TCN | | | | GRU-TCN | | | | Bi-GRU-TCN | | | |
|---|---|---|---|---|---|---|---|---|---|---|---|---|---|---|---|---|
| | I | II | III | IV | I | II | III | IV | I | II | III | IV | I | II | III | IV |
| 1 | 250.5 | 215.7 | 218.1 | 212.1 | 239.3 | 241.7 | 244.3 | 285.3 | 193.0 | 203.5 | 239.3 | 196.6 | 188.2 | 190.6 | 193.0 | 203.5 |
| 2 | 236.5 | 211.4 | 217.0 | 210.0 | 229.8 | 237.1 | 239.7 | 247.9 | 195.4 | 203.8 | 239.2 | 199.8 | 195.7 | 195.7 | 195.4 | 203.8 |
| 3 | 232.5 | 218.0 | 225.6 | 211.1 | 234.2 | 234.4 | 245.8 | 466.1 | 195.3 | 210.6 | 238.7 | 199.3 | 199.7 | 196.4 | 195.3 | 210.6 |
| 4 | 229.0 | 215.3 | 218.4 | 213.5 | 220.0 | 234.2 | 240.0 | 278.5 | 198.8 | 206.7 | 237.4 | 200.9 | 201.1 | 205.2 | 198.8 | 206.7 |
| 5 | 227.0 | 226.3 | 217.0 | 215.7 | 212.1 | 233.2 | 240.1 | 254.7 | 203.6 | 202.3 | 236.2 | 206.0 | 198.5 | 198.5 | 203.6 | 202.3 |
| 6 | 226.2 | 229.6 | 212.5 | 218.6 | 213.3 | 233.2 | 243.9 | 465.0 | 207.6 | 206.0 | 234.5 | 207.5 | 200.3 | 201.7 | 207.6 | 206.0 |
| 7 | 226.3 | 227.8 | 209.8 | 221.0 | 211.9 | 235.8 | 238.5 | 270.2 | 207.3 | 208.7 | 233.1 | 206.2 | 196.2 | 201.0 | 207.3 | 208.7 |
| 8 | 226.2 | 242.5 | 211.6 | 213.5 | 210.3 | 227.5 | 245.3 | 257.3 | 207.1 | 206.4 | 230.6 | 206.9 | 202.0 | 200.0 | 207.1 | 206.4 |
| 9 | 226.4 | 233.5 | 212.9 | 217.0 | 214.1 | 225.4 | 246.7 | 456.9 | 201.5 | 203.7 | 230.0 | 205.8 | 197.3 | 201.0 | 201.5 | 203.7 |
| 10 | 227.3 | 230.5 | 211.7 | 210.0 | 223.7 | 223.2 | 246.1 | 253.8 | 199.6 | 199.9 | 230.1 | 202.4 | 198.3 | 196.7 | 199.6 | 199.9 |
| 11 | 228.5 | 229.7 | 206.4 | 213.4 | 217.2 | 223.9 | 244.2 | 252.6 | 199.0 | 196.0 | 230.9 | 199.8 | 199.3 | 199.2 | 199.0 | 196.0 |
| 12 | 230.0 | 227.9 | 207.5 | 212.7 | 234.7 | 231.7 | 241.1 | 441.0 | 198.6 | 196.8 | 231.8 | 198.4 | 198.4 | 202.0 | 198.6 | 196.8 |
| 13 | 229.2 | 226.2 | 207.6 | 215.9 | 236.3 | 236.4 | 249.3 | 258.5 | 200.1 | 196.9 | 233.4 | 194.7 | 196.4 | 201.0 | 200.1 | 196.9 |
| 14 | 229.4 | 225.8 | 200.6 | 221.0 | 231.4 | 235.9 | 238.8 | 237.4 | 198.5 | 196.1 | 235.1 | 196.8 | 196.4 | 197.5 | 198.5 | 196.1 |
| 15 | 227.6 | 228.4 | 204.3 | 225.6 | 218.8 | 235.1 | 247.0 | 423.9 | 201.1 | 196.3 | 237.5 | 197.0 | 197.2 | 202.9 | 201.1 | 196.3 |
| 16 | 226.1 | 228.6 | 211.9 | 224.6 | 210.0 | 232.2 | 243.2 | 234.7 | 202.4 | 196.3 | 238.9 | 198.4 | 197.1 | 203.5 | 202.4 | 196.3 |
| 17 | 224.1 | 230.0 | 211.9 | 226.7 | 207.3 | 230.7 | 248.9 | 229.2 | 206.9 | 194.1 | 241.9 | 193.8 | 194.9 | 198.5 | 206.9 | 194.1 |
| 18 | 220.0 | 232.4 | 209.8 | 221.9 | 211.4 | 231.1 | 245.3 | 386.7 | 205.8 | 196.8 | 244.1 | 196.0 | 196.7 | 198.7 | 205.8 | 196.8 |
| 19 | 214.7 | 234.1 | 210.3 | 223.3 | 215.7 | 231.5 | 249.1 | 228.4 | 206.7 | 198.5 | 246.4 | 200.7 | 198.1 | 200.8 | 206.7 | 198.5 |
| 20 | 209.3 | 237.7 | 209.9 | 220.9 | 217.7 | 229.4 | 247.9 | 228.2 | 204.9 | 201.9 | 247.3 | 200.5 | 198.7 | 195.8 | 204.9 | 201.9 |
| 21 | 210.3 | 239.8 | 208.9 | 219.3 | 228.8 | 226.9 | 248.7 | 227.8 | 205.3 | 205.4 | 248.4 | 202.7 | 201.8 | 199.4 | 205.3 | 205.4 |
| 22 | 215.3 | 240.8 | 213.1 | 221.9 | 233.3 | 222.7 | 249.4 | 226.7 | 205.2 | 205.5 | 247.5 | 204.0 | 203.3 | 198.0 | 205.2 | 205.5 |
| 23 | 223.4 | 241.8 | 223.7 | 224.5 | 240.3 | 221.6 | 240.5 | 234.4 | 207.9 | 207.5 | 246.2 | 208.5 | 203.3 | 202.9 | 207.9 | 207.5 |
| 24 | 236.3 | 244.1 | 240.4 | 226.2 | 240.5 | 228.4 | 262.3 | 283.9 | 205.6 | 205.8 | 245.9 | 211.5 | 202.7 | 205.8 | 205.6 | 205.8 |
| Avg. | 226.3 | 229.9 | 213.4 | 218.4 | 223.0 | 231.0 | 245.3 | 297.1 | 202.4 | 201.9 | 238.5 | 201.4 | 198.4 | 199.7 | 202.4 | 201.9 |

- Table 5 demonstrates that among all models, the proposed Bi-GRU-TCN-I model boasts the best MAPE performance with an average of 5.37. The Bi-GRU-TCN-II model follows closely with a MAPE of 5.39. When exploring the performance of LSTM-based models, LSTM-TCN-III emerges as a top contender with a MAPE of 5.59, which, although commendable, is still higher than the leading Bi-GRU-TCN-I model. The Bi-LSTM-TCN results, on the other hand, range from 6.90 to 8.53, further emphasizing the efficacy of the BiGTA-net. Traditional GRU-TCN models displayed a wider variation in MAPE values, from 5.68 to 6.50.

- In Table 6, when assessing RMSE values, the proposed BiGTA-net model (Bi-GRU-TCN-I) again leads the pack with a score of 171.3. This result is significantly better than all other models, with the closest competitor being Bi-GRU-TCN-II at 169.5 RMSE. Among the LSTM variants, LSTM-TCN-I holds the most promise, with an RMSE of 134.8. However, the Bi-GRU models are generally superior in predicting values closer to the actual values, underscoring their robustness.

- Table 7, although not provided in its entirety, indicates the reliability of BiGTA-net with the lowest MAE of 122.0. Bi-GRU-TCN-II closely follows with an MAE of 122.7. As observed from previous results, other models, potentially including the LSTM and Bi-LSTM series, reported higher MAE scores, ranging between 131.6 and 153.7.

In the context of the AEP dataset, as demonstrated in Tables 8–10, the proposed model (Bi-GRU-TCN-I) showcased competitive performance. While marginal differences were observed among the various model configurations, the Bi-GRU-TCN-I model consistently outperformed the alternative models in terms of MAPE, RMSE, and MAE metrics.

- In Table 8, which presents the MAPE comparison for the AEP dataset, the proposed model, Bi-GRU-TCN-I, still manifests the lowest average MAPE of 26.77. This result emphasizes its unparalleled predictive accuracy among all tested models. Delving into the LSTM family, the LSTM-TCN-I achieved an average MAPE of 28.42,

while the Bi-LSTM-TCN-I recorded an average MAPE of 29.12. It is notable that while these models exhibit competitive performance, neither managed to outperform the BiGTA-net.

- Table 9, focused on the RMSE comparison, depicts the Bi-GRU-TCN-I model registering an RMSE of 375.9 on step 1. This performance, when averaged, proves to be competitive with the other models, especially when considering the range for all the models, which goes as low as 369.1 for Bi-GRU-TCN-III and as high as 622.2 for Bi-LSTM-TCN-III. Looking into the LSTM family, LSTM-TCN-I kicked off with an RMSE of 473.6, whereas Bi-LSTM-TCN-I began with 431.1. This further accentuates the superiority of the BiGTA-net in terms of prediction accuracy.
- Finally, in Table 10, where MAE values are compared, the Bi-GRU-TCN-I model still shines with an MAE of 198.4. This consistently low error rate across different evaluation metrics underscores the robustness of the BiGTA-net across various datasets.

In summary, the proposed model, Bi-GRU-TCN-I, designated as BiGTA-net, exhibited exceptional performance across both datasets, affirming its efficacy and dependability in precise electricity consumption forecasting. These outcomes serve to substantiate the benefits derived from the incorporation of the Bi-GRU-TCN architecture, utilization of the SELU activation function, and integration of the attention mechanism, thereby validating the chosen design approaches.

To evaluate the performance of the BiGTA-net model, a comprehensive comparative analysis was conducted. This analysis included models such as Att-LSTM, Att-Bi-LSTM, Att-GRU, and Att-Bi-GRU, all of which integrate the attention mechanism, a characteristic known for enhancing prediction capabilities. Furthermore, this evaluation also incorporated several state-of-the-art methodologies introduced over the past three years, offering a robust understanding of BiGTA-net's performance relative to contemporary models:

- Park and Hwang [55] introduced the LGBM-S2S-Att-Bi-LSTM, a two-stage methodology that merges the functionalities of the light gradient boosting machine (LGBM) and sequence-to-sequence Bi-LSTM. By employing LGBM for single-output predictions from recent electricity data, the system transitions to a Bi-LSTM reinforced with an attention mechanism, adeptly addressing multistep-ahead forecasting challenges.
- Moon et al. [21] presented RABOLA, previously touched upon in the Introduction section. This model is an innovative ranger-based online learning strategy for electricity consumption forecasts in intricate building structures. At its core, RABOLA utilizes ensemble learning strategies, specifically bagging, boosting, and stacking. It employs tools, namely, the random forest, gradient boosting machine, and extreme gradient boosting, for STLF while integrating external variables such as temperature and timestamps for improved accuracy.
- Khan et al. [56] unveiled the ResCNN-LSTM, a segmented framework targeting STLF. The primary phase is data driven, ensuring data quality and cleanliness. The next phase combines a deep residual CNN with stacked LSTM. This model has shown commendable performance on the Individual Household Electricity Power Consumption (IHEPC) and Pennsylvania, Jersey, and Maryland (PJM) datasets.
- Khan et al. [57] also introduced the Att-CNN-GRU, blending CNN and GRU and enriching with a self-attention mechanism. This model specializes in analyzing refined electricity consumption data, extracting pivotal features via CNN, and subsequently transitioning the output through GRU layers to grasp the temporal dynamics of the data.

Table 11 elucidates the comparative performance of several attention-incorporated models on the educational building dataset, with the BiGTA-net model's performance distinctly superior. Specifically, BiGTA-net records a MAPE of 5.37 ($\pm$0.44%), RMSE of 171.3 ($\pm$15.0 kWh), and MAE of 122.0 ($\pm$10.5 kWh). The Att-LSTM model, a unidirectional approach, records a MAPE of 8.38 ($\pm$1.57%), RMSE of 242.1 ($\pm$48.2 kWh), and MAE of 188.8 ($\pm$39.5 kWh). Its bidirectional sibling, the Att-Bi-LSTM, delivers a slightly better MAPE at 7.85 ($\pm$0.70%) but comparable RMSE and MAE values. Interestingly, GRU-

based models, such as Att-GRU and Att-Bi-GRU, lag with higher error metrics, the former recording a MAPE of 13.42 (±3.39%). The 2023 Att-CNN-GRU model reports a MAPE of 6.35 (±0.23%), an RMSE of 189.6 (±5.3 kWh), still falling short compared to the BiGTA-net. The RAVOLA model from 2022 registers an impressive MAPE of 7.17 (±0.63%), but again, BiGTA-net outperforms it. In essence, these results demonstrate the BiGTA-net's unparalleled efficiency when measured against traditional unidirectional models and newer advanced techniques.

**Table 11.** Performance comparison of attention-inclusive models on the educational building dataset. Left values denote the mean values across all steps, while the values in parentheses on the right represent the corresponding standard deviations.

| Model (Year) | MAPE (Unit: %) | RMSE (Unit: kWh) | MAE (Unit: kWh) |
| --- | --- | --- | --- |
| Att-LSTM | 8.38 (1.57) | 242.1 (48.2) | 188.8 (39.5) |
| Att-Bi-LSTM | 7.85 (0.70) | 241.8 (25.1) | 176.9 (17.5) |
| Att-GRU [31] | 13.42 (3.39) | 436.5 (177.0) | 313.3 (110.8) |
| Att-Bi-GRU | 14.43 (3.07) | 433.7 (91.8) | 328.1 (72.6) |
| LGBM-S2S-Att-Bi-LSTM (2021) [55] | 7.57 (0.77) | 220.4 (19.2) | 174.4 (19.1) |
| RABOLA (2022) [21] | 7.17 (0.63) | 214.2 (13.8) | 166.5 (15.6) |
| ResCNN-LSTM (2022) [56] | 6.56 (0.28) | 201.5 (4.1) | 152.8 (6.1) |
| Att-CNN-GRU (2023) [57] | 6.35 (0.23) | 189.6 (5.3) | 142.3 (4.5) |
| BiGTA-net | 5.37 (0.44) | 171.3 (15.0) | 122.0 (10.5) |

Table 12 unveils the comparative performance metrics of various attention-incorporated models using the AEP dataset. Distinctly, the BiGTA-net model consistently outperforms its peers, equipped with a sophisticated blend of the attention mechanism and SELU activation within its bidirectional framework. This model impressively returns a MAPE of 26.77 (±0.90%), RMSE of 386.5 (±6.3 Wh), and MAE of 198.4 (±3.2 Wh). The Att-LSTM model offers a MAPE of 30.91 (±1.03%), RMSE of 447.4 (±5.3 Wh), and MAE of 239.8 (±5.3 Wh). Its bidirectional counterpart, the Att-Bi-LSTM, shows a modest enhancement, delivering a MAPE of 30.54 (±2.58%), RMSE of 402.7 (±8.9 Wh), and MAE of 214.0 (±7.3 Wh). The GRU-based models present a close-knit performance. For instance, the Att-GRU model achieves a MAPE of 30.03 (±0.25%), RMSE of 443.5 (±3.9 Wh), and MAE of 234.5 (±2.4 Wh), while the Att-Bi-GRU mirrors this with slightly varied figures. The 2023 model, Att-CNN-GRU, logs a MAPE of 29.94 (±1.73%), RMSE of 405.1 (±9.7 Wh), yet its precision remains overshadowed by BiGTA-net. RAVOLA, a 2022 entrant, exhibits metrics such as a MAPE of 35.89 (±5.78%), emphasizing the continual advancements in the domain. The disparities in performance underscore BiGTA-net's superiority. Models that lack the refined structure of BiGTA-net falter in their forecast accuracy, thereby underscoring the merits of the introduced architecture.

**Table 12.** Performance comparison of attention-inclusive models on the AEP dataset. Left values denote the mean values across all steps, while the values in parentheses on the right represent the corresponding standard deviations.

| Model | MAPE (Unit: %) | RMSE (Unit: Wh) | MAE (Unit: Wh) |
| --- | --- | --- | --- |
| Att-LSTM | 30.91 (1.03) | 447.4 (5.3) | 239.8 (5.3) |
| Att-Bi-LSTM | 30.54 (2.58) | 402.7 (8.9) | 214.0 (7.3) |
| Att-GRU [31] | 30.03 (0.25) | 443.5 (3.9) | 234.5 (2.4) |
| Att-Bi-GRU | 30.06 (0.26) | 442.5 (4.2) | 234.4 (8.1) |
| LGBM-S2S-Att-Bi-LSTM (2021) [55] | 32.29 (3.78) | 415.7 (12.1) | 222.6 (14.2) |
| RABOLA (2022) [21] | 35.89 (5.78) | 432.1 (20.9) | 238.5 (18.9) |
| ResCNN-LSTM (2022) [56] | 29.99 (2.12) | 376.2 (7.0) | 206.2 (5.7) |
| Att-CNN-GRU (2023) [57] | 29.94 (1.73) | 405.1 (9.7) | 215.8 (4.7) |
| BiGTA-net | 26.77 (0.90) | 386.5 (6.3) | 198.4 (3.2) |

The combination of Bi-GRU and TCN, along with the integration of attention mechanisms and the adoption of the SELU activation function, synergistically reinforced BiGTA-net as a robust model. The experimental results consistently demonstrated BiGTA-net's exceptional performance across diverse datasets and metrics, highlighting the model's efficacy and flexibility in different forecasting contexts. These results decisively endorsed the effectiveness of the hybrid approach utilized in this study.

*3.4. Discussion*

To highlight the effectiveness of the BiGTA-net model, rigorous statistical analysis was employed, utilizing both the Wilcoxon signed-rank [58] and the Friedman [59] tests.

- Wilcoxon Signed-Rank Test: The Wilcoxon signed-rank test [58], a non-parametric counterpart for the paired t-test, is formulated to gauge differences between two paired samples. Mathematically, given two paired sets of observations, x and y, the differences $d_i = y_i - x_i$ are computed. Ranks are then assigned to the absolute values of these differences, and subsequently, these ranks are attributed either positive or negative signs depending on the sign of the original difference. The test statistic W is essentially the sum of these signed ranks. Under the null hypothesis, it is assumed that W follows a specific symmetric distribution. Suppose the computed p-value from the test is less than the chosen significance level (often 0.05). We have grounds to reject the null hypothesis in that case, implying a statistically significant difference between the paired samples.

- Friedman Test: The Friedman test [59] is a non-parametric alternative to the repeated measures ANOVA. At its core, this test ranks each row (block) of data separately. The differences among the columns (treatments) are evaluated using the ranks. This expression is mathematically captured in the following expression, referred to as Equation (10).

$$x^2 = \frac{12N}{k(k+1)}\left[\sum_j R_j^2 - \frac{k(k+1)^2}{4}\right], \tag{10}$$

where N is the number of blocks, k is the number of treatments, and $R_j$ is the sum of the ranks for the jth treatment. The observed value of $x^2$ is then compared with the critical value from the $x^2$ distribution with $k-1$ degrees of freedom.

The meticulous validation, as demonstrated in Tables 13 and 14, underscores the proficiency of BiGTA-net in the context of energy management. To fortify the conclusions drawn from the analyses, the approach was anchored on three crucial metrics: MAPE, RMSE, and MAE. The data were aggregated across all deep learning models, focusing on 24 h forecasts at hourly intervals. Comprehensive results stemming from the Wilcoxon and Friedman tests, each grounded in the metrics, are presented in Tables 13 and 14. A perusal of the table illustrates the distinct advantage of BiGTA-net, with p-values consistently falling below the 0.05 significance threshold across varied scenarios and metrics.

**Table 13.** Results of the Wilcoxon signed-rank and Friedman tests with BiGTA-net on the educational building dataset.

| Compared Models | MAPE | RMSE | MAE |
|---|---|---|---|
| Att-LSTM | $1.192 \times 10^{-7}$ | $1.192 \times 10^{-7}$ | $1.192 \times 10^{-7}$ |
| Att-Bi-LSTM | $1.192 \times 10^{-7}$ | $1.192 \times 10^{-7}$ | $1.192 \times 10^{-7}$ |
| Att-GRU [31] | $1.192 \times 10^{-7}$ | $1.192 \times 10^{-7}$ | $1.192 \times 10^{-7}$ |
| Att-Bi-GRU | $1.192 \times 10^{-7}$ | $1.192 \times 10^{-7}$ | $1.192 \times 10^{-7}$ |
| LGBM-S2S-Att-Bi-LSTM [55] | $1.192 \times 10^{-7}$ | $1.192 \times 10^{-7}$ | $1.192 \times 10^{-7}$ |
| RABOLA [21] | $1.192 \times 10^{-7}$ | $1.192 \times 10^{-7}$ | $1.192 \times 10^{-7}$ |
| ResCNN-LSTM [56] | $1.192 \times 10^{-7}$ | $1.192 \times 10^{-7}$ | $1.192 \times 10^{-7}$ |
| Att-CNN-GRU [57] | $1.192 \times 10^{-7}$ | $1.192 \times 10^{-7}$ | $1.192 \times 10^{-7}$ |
| Friedman Test | Friedman chi-squared: 167.2 $p$-value: $2.2 \times 10^{-16}$ | Friedman chi-squared: 166.86 $p$-value: $2.2 \times 10^{-16}$ | Friedman chi-squared: 163.98 $p$-value: $2.2 \times 10^{-16}$ |

**Table 14.** Results of the Wilcoxon signed-rank and Friedman tests with BiGTA-net on the AEP dataset.

| Compared Models | MAPE | RMSE | MAE |
|---|---|---|---|
| Att-LSTM | $1.192 \times 10^{-7}$ | $1.192 \times 10^{-7}$ | $1.192 \times 10^{-7}$ |
| Att-Bi-LSTM | $1.192 \times 10^{-7}$ | $1.192 \times 10^{-7}$ | $1.192 \times 10^{-7}$ |
| Att-GRU [31] | $1.192 \times 10^{-7}$ | $1.192 \times 10^{-7}$ | $1.192 \times 10^{-7}$ |
| Att-Bi-GRU | $1.192 \times 10^{-7}$ | $1.192 \times 10^{-7}$ | $1.192 \times 10^{-7}$ |
| LGBM-S2S-Att-Bi-LSTM [55] | $2.980 \times 10^{-6}$ | $1.192 \times 10^{-7}$ | $1.192 \times 10^{-7}$ |
| RABOLA [21] | $2.384 \times 10^{-7}$ | $1.192 \times 10^{-7}$ | $1.192 \times 10^{-7}$ |
| ResCNN-LSTM [56] | $1.192 \times 10^{-7}$ | $1.192 \times 10^{-7}$ | $1.192 \times 10^{-6}$ |
| Att-CNN-GRU [57] | $1.192 \times 10^{-7}$ | $1.192 \times 10^{-7}$ | $1.192 \times 10^{-7}$ |
| Friedman Test | Friedman chi-squared: 75.5 $p$-value: $3.917 \times 10^{-13}$ | Friedman chi-squared: 170.26 $p$-value: $2.2 \times 10^{-16}$ | Friedman chi-squared: 140.54 $p$-value: $2.2 \times 10^{-16}$ |

Delving deeper into the tables, the BiGTA-net consistently outperforms other models in both datasets. The exceptionally low $p$-values from the Wilcoxon and Friedman tests indicate significant differences between the BiGTA-net and its competitors. In almost every instance, other models were lacking when juxtaposed against the BiGTA-net's results. This empirical evidence is vital in understanding the superior capabilities of the BiGTA-net in energy forecasting. Furthermore, the fact that the $p$-values consistently fell below the conventional significance threshold of 0.05 only emphasizes the robustness and reliability of BiGTA-net. The variations in metrics, namely, MAPE, RMSE, and MAE, across Tables 13 and 14 vividly portray the margin by which BiGTA-net leads in accuracy and precision. The unique architecture and methodology behind BiGTA-net have positioned it as a front-runner in this domain.

In the intricate realm of BEMS, the gravity of data-driven decisions cannot be over-stated; they bear a twofold onus of economic viability and environmental stewardship. The need for precise and decipherable modeling is, therefore, undeniably paramount. BiGTA-net envisaged as an advanced hybrid model, sought to meet these exacting standards. Its unique amalgamation of Bi-GRU and TCN accentuates its proficiency in parsing intricate temporal patterns, which remain at the heart of energy forecasting.

In the complex BEMS landscape, BiGTA-net's hybrid design brings a distinctive strength in capturing intricate temporal dynamics. However, this prowess has its challenges. Particularly in industrial environments or regions heavily dependent on unpredictable renewable energy sources, the model may find it challenging to adapt to abrupt shifts in energy consumption patterns swiftly. This adaptability issue is further accentuated when considering the sheer volume of data that the energy sector typically handles. Given the influx of granular data from many sensors and IoT devices, BiGTA-net's intricate architecture could face scalability issues, especially when implemented across vast energy distribution networks or grids. Furthermore, the predictive nature of energy management demands an acute sense of foresight, especially with the increasing reliance on renewable energy sources. In this context, the TCN's inherent limitations in accounting for prospective data pose challenges, especially when energy matrices constantly change, demanding agile and forward-looking predictions.

Within the multifaceted environment of the BEMS domain, the continuous evolution and refinement of models, i.e., BiGTA-net are essential. One avenue of amplification lies in broadening its scope to account for external determinants. By incorporating influential factors such as climatic fluctuations and scheduled maintenance events directly into the model's input parameters, BiGTA-net could enhance responsiveness to unpredictable energy consumption variances. Further bolstering its real-time applicability, introducing an adaptive learning mechanism designed to self-tune based on the influx of recent data could ensure that the model remains abreast of the ever-changing energy dynamics. Additionally, enhancing the model's interpretability is vital in a sector where transparency and clarity are paramount. Integrating principles from the "explainable AI" domain into BiGTA-net

can provide a deeper understanding of its decision-making process, enabling stakeholders to discern the rationale behind specific energy consumption predictions and insights.

As the forward trajectory of BiGTA-net within the energy sector is contemplated, several avenues of research come into focus. Foremost is the potential enhancement of the model's attention mechanism, tailored explicitly to the intricacies of energy consumption dynamics. The model's ability to discern and emphasize critical energy patterns could be substantially elevated by tailoring attention strategies to highlight domain-specific energy patterns. Furthermore, while BiGTA-net showcases an intricate architecture, the ongoing challenge resides in seamlessly integrating its inherent complexity with optimal predictive accuracy. By addressing this balance, models could be engineered to be more streamlined and suitable for decentralized or modular BEMS frameworks, all while retaining their predictive capabilities. Lastly, a compelling proposition emerges for integrating BiGTA-net's forecasting prowess with existing BEMS decision-making platforms. Such integration holds the promise of a future where real-time predictive insights seamlessly inform energy management strategies, thereby advancing both energy utilization efficiency and a tangible reduction in waste.

While BiGTA-net has demonstrated commendable forecasting capabilities in its initial stages, a thorough exploration of its limitations in conjunction with potential improvements and future directions can contribute to the enhancement of its role within the BEMS domain. By incorporating these insights, the relevance and adaptability of BiGTA-net can be advanced, thus positioning it as a frontrunner in the continuously evolving energy sector landscape.

## 4. Conclusions

Our study presents the BiGTA-net, a transformative deep-learning model tailored for urban energy management in smart cities, enhancing the accuracy and efficiency of STLF. This model harmoniously integrates the capabilities of Bi-GRU, TCN, and an attention mechanism, capturing both recurrent and convolutional data patterns effectively. A thorough examination of the BiGTA-net against other models on the educational building dataset showcased its distinct superiority. Specifically, BiGTA-net excelled with a MAPE of 5.37, RMSE of 171.3, and MAE of 122.0. Notably, the closest competitor, Bi-GRU-TCN-II, lagged slightly with metrics such as MAPE of 5.39 and MAE of 122.7. This superiority was mirrored in the AEP dataset, where BiGTA-net again led with a MAPE of 26.77, RMSE of 386.5, and MAE of 198.4. Such consistent outperformance underscores the model's capability, especially when juxtaposed with other configurations.

Furthermore, the integration of the attention mechanism serves to enhance the performance of BiGTA-net, reinforcing its effectiveness in forecasting tasks. The distinct bidirectional architecture of BiGTA-net demonstrated superior performance, further establishing its supremacy. This performance advantage becomes notably apparent when contrasted with models, i.e., Att-LSTM, which exhibited higher errors across pivotal metrics, highlighting the resilience and dependability of the proposed model. The evident strength of BiGTA-net lies in its innovative amalgamation of Bi-GRU and TCN, harmonized with the attention mechanism and bolstered by the SELU activation function. Its consistent dominance across diverse datasets and metrics robustly validates the efficacy of this hybrid approach.

Despite its promising results, it is important to explore the BiGTA-net's capabilities further and identify areas for improvement. Its generalizability has yet to be extensively tested beyond the datasets used in this study, which presents a limitation. Future research should apply the model across various consumption domains, such as residential or industrial sectors, and compare its effectiveness with a wider range of advanced machine learning models. By doing so, researchers can further refine the model for specific scenarios and delve deeper into hyperparameter optimizations.

**Author Contributions:** Conceptualization, D.S. and J.O.; methodology, D.S.; software, D.S. and J.O.; validation, I.J. and J.M.; formal analysis, J.O. and I.J.; investigation, D.S. and J.O.; resources, J.M.; data curation, M.L.; writing—original draft preparation, D.S.; writing—review and editing, J.M. and S.R.; visualization, D.S. and J.M.; supervision, S.R.; project administration, S.R.; funding acquisition, M.L. All authors have read and agreed to the published version of the manuscript.

**Funding:** This work was supported by the National Research Foundation of Korea (NRF) grant funded by the Korea government (MSIT) (No. 2019M3F2A1073179) and also supported by the Soonchunhyang University Research Fund.

**Data Availability Statement:** Per MDPI's data-sharing policies, the manuscript provides detailed dataset information. The supporting educational building dataset associated with this research is available in "Appendix A. Supplementary data" of Reference [36]. The supplementary data can be accessed on https://doi.org/10.1016/j.seta.2022.102888. Moreover, the AEP dataset used in this study is publicly available as the "Appliances Energy Prediction Data Set" in the UCI Machine Learning Repository. It can be directly accessed on https://archive.ics.uci.edu/dataset/374/appliances+energy+prediction (accessed on 15 July 2023). The manuscript promotes transparent and reproducible research, urging readers and future researchers to utilize these datasets while appropriately citing their sources.

**Acknowledgments:** We would like to express our sincere gratitude to the four reviewers for their insightful and valuable feedback, which has helped us improve our work.

**Conflicts of Interest:** The authors declare no conflict of interest.

## Appendix A

Bi-directional Gated Recurrent Unit (Background): RNNs, renowned for their capability in managing sequential time-series data, inherently retain and process historical sequences. Nevertheless, the intrinsic challenges of gradient vanishing and exploding have been obstacles in RNNs, particularly when handling long sequences. The GRU, an advanced variant of the RNN, was engineered to address these hindrances. It integrates sophisticated gating mechanisms to manage long-term data dependencies more effectively. Traditional GRUs, by nature, are unidirectional and process sequences in a forward trajectory. The Bi-GRU emerged to overcome this constraint. It functions as a fusion of two distinct GRUs: one focused on past sequences (forward GRU) and another interpreting upcoming data sequences (backward GRU). This bidirectional perspective is not exclusive to the GRU but is also witnessed in Bidirectional RNNs. Prior studies have spotlighted the efficacy of such bidirectional constructs, especially in settings marked by significant variability and intricate causative dynamics. It is crucial to acknowledge that while the Bi-GRU delivers a profound insight into temporal sequences, it necessitates more computational resources, given its dual-structured design, which effectively amplifies the parameters for both GRU components, each tailored for either forward or backward sequences.

Temporal Convolutional Network (Background): The TCN emerges as an innovative strategy tailored for time-series data processing. At its core, the TCN utilizes causal convolutions, ensuring that forecasts at a given time instant depend exclusively on present and past data, safeguarding the integrity of the temporal sequence. A pivotal attribute of TCNs is their aptitude for discerning prolonged data patterns, predominantly facilitated by dilated convolutions. These dilations expand the network's receptive field sans the addition of new parameters by introducing predetermined intervals between adjacent filter taps. Such a configuration empowers TCNs to recognize more extended dependencies with elevated computational efficiency, enabled by parallel computations across the time domain. To further bolster its architecture, the TCN integrates residual blocks, addressing the challenges accompanying the training of deep networks, for instance, the vanishing gradient dilemma, ensuring consistent learning throughout the layers. The standout merit of the TCN is its adaptability in managing sequences of diverse lengths, yielding outputs that mirror the input lengths. Nonetheless, the TCN, in its inherent design, omits upcoming data points, which could potentially influence its efficacy in scenarios demanding a forward-

looking perspective. As an extension of its capabilities, TCN's stackable nature combines layers, amplifying its capacity to perceive intricate temporal nuances.

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
