# Peer review of "BiGTA-Net: A Hybrid Deep Learning-Based Electrical Energy Forecasting Model for Building Energy Management Systems"

_systems, doi:10.3390/systems11090456_

Round 1

Reviewer 1 Report

In this study, the researchers have integrated the Bidirectional Gated Recurrent Unit (Bi-GRU), Temporal Convolutional Network (TCN), and Attention Mechanism to enhance their approach. However, it appears that the study does not introduce entirely novel methodologies. As a result, the contributions of this study could be considered somewhat constrained.

Several points have been identified that warrant consideration:

1. Regarding the implementation of GRU using Keras in the experiments, it's worth noting that Equation (7) related to GRU seems to contain inaccuracies. A recommendation is made to verify Equation (7) in the Keras open-source documentation.

2. Section 4 might be better utilized for elucidating original concepts. It is observed that the concepts of the Bidirectional Gated Recurrent Unit (Bi-GRU), Temporal Convolutional Network (TCN), and Attention Mechanism were not originated by the authors and could potentially be excluded from Section 4. These elements could potentially find a suitable place in a background section.

3. The equations presented in Section 4 are not inherently proposed by the authors. Thus, a suggestion is made to consider removing all equations from Section 4.

4. It is advised that the authors delve into comprehensive explication of their original concepts and undertake a comparative analysis with existing methodologies.

5. While the comparison between GRU-TCN and Bi-GRU-TCN has been addressed, it could be beneficial to expand this comparison to encompass other configurations, such as LSTM-TCN, Bi-LSTM-TCN, LSTM-TCN-Attention Mechanism, and Bi-LSTM-TCN-Attention Mechanism.

6. It is suggested that the authors broaden their comparative analysis to encompass methodologies from esteemed journals and conferences published within the past three years.

7. Consideration should be given to furnishing original mathematical models that substantiate the efficacy of the proposed approach. Additionally, inclusion of relevant theoretical frameworks would provide valuable support for the proposed methodology.

8. It would be prudent to engage in a discussion regarding the limitations inherent in the proposed methodology within Section 6. This would contribute to a well-rounded understanding of the study's scope and potential constraints.

Author Response

We would like to extend our sincerest gratitude to Reviewer 1 for your insightful and detailed review. Your comments have been invaluable in improving our work, and we sincerely appreciate the time and effort you have put in to evaluate our submission. Please find the corrected file and our response attached, and thank you again for your support.

Reviewer 2 Report

The authors are proposing a new system for energy load forecasting. The approach and the results are interesting but the paper is missing several important components:

1. Explain how the values for the constants in equations 5 and 6 were found.

2. In equations 1-4, explain what you mean by x and y subscripts. Why do you need 4 equations instead of two?

3. You should add few comparisons with other proposed methods in the literature. 

4. Literature review is focused on traditional deep learning methods and missing recent pape Ers that use transformers and hypercomplex neural networks. Please cite the following related paper:

Wind Speed Forecasting Using the Stationary Wavelet Transform and Quaternion Adaptive-Gradient Methods

Author Response

We would like to extend our sincerest gratitude to Reviewer 2 for your insightful and detailed review. Your comments have been invaluable in improving our work, and we sincerely appreciate the time and effort you have put in to evaluate our submission. Please find the corrected file and our response attached, and thank you again for your support.

Reviewer 3 Report

The paper entitled BiGTA-Net: A Hybrid Deep Learning-Based Electrical Energy Forecasting Model for Building Energy Management Systems introduces a hybrid load prediction method by combining BiGRU, TCN and ATT. The paper is well-written with good figures and nice tables, here are some suggestions which can improve this manuscript:

1. The frontsize of the figures is too small and it is hard to tell the text, especially Figure 2 and 3.

2. The literature review part does not cover the state-of-the-art work in the field of wavelt transform-based methods, such as:

[1] Wang, Y.; Guo, P.; Ma, N.; Liu, G. Robust Wavelet Transform Neural-Network-Based Short-Term Load Forecasting for Power Distribution Networks. Sustainability 2023, 15, 296.

[2] X. Zhang, S. Kuenzel, N. Colombo and C. Watkins, "Hybrid Short-term Load Forecasting Method Based on Empirical Wavelet Transform and Bidirectional Long Short-term Memory Neural Networks," in Journal of Modern Power Systems and Clean Energy, vol. 10, no. 5, pp. 1216-1228, September 2022

Also, the reviewer is curious about the performance of the proposed method against wavelet decomposition methods, the authors are expected to add a group of simulation to make a comparision.

3. The English must be improved.

4. The conlusion part is too wordy, try to use concise sentence to summarize your work.

The English should be proofread by native speaker

Author Response

We would like to extend our sincerest gratitude to Reviewer 3 for your insightful and detailed review. Your comments have been invaluable in improving our work, and we sincerely appreciate the time and effort you have put in to evaluate our submission. Please find the corrected file and our response attached, and thank you again for your support.

Reviewer 4 Report

Review of  BiGTA-Net: A Hybrid Deep Learning-Based Electrical Energy  Forecasting Model for Building Energy Management Systems

This manuscript presents  hybrid deep learning model that blends a bi-directional gated recurrent unit (Bi-GRU), a temporal convolutional network (TCN), and an attention mechanism. BiGTA-net circumvents the limitations of traditional deep learning models, offering an innovative and efficient solution for STLF.

This manuscript provides some interesting information, however revision is recommended before it may be considered future, namely:

1. please also underline the novelty of this research in abstract;

2. please write the manuscript in third person --- please avoid …we… --- please revise the manuscript;

3. please provide some most important measurable results in abstract;

4. in introduction please clearly where the energy is used (heating,  DHW, and/or ….) --- because this is not clear for the reader;

5. At the end of introduction please underline the novelty of this research and archival value of results, if it may be published in high quality journal AND please do not provide summary of the manuscript in this section;

6. The main objective of the paper must be written on the clearer and more concise way at the end of introduction section;

7. Please be aware that using multiple references is not very helpful for a reader. If authors need to use more references at least a short assessment/justification should be provided;

8. Introduction section must be written on more quality way, i.e. more up-to-date references addressed. Research gap should be delivered on more clear way with directed necessity for the conducted research work. Please see for example:

https://doi.org/10.1016/j.jobe.2018.07.021
https://doi.org/10.1016/j.enbuild.2020.109831

https://doi.org/10.1016/j.jclepro.2022.131605

9. it is suggested to integrate section 1 and 2;

10. in present section 3: please clearly state what for the forecast energy is used;

11. please do not mix materials and method with result and discussion --- now it is mix what is not clear for the reader. Please try to have materials and method AND results and discussion sections;

12. please provide block schema with your method in order to show main inputs and outputs + main innovations;

  1. Critical discussion about the applicability and weak points of the proposed approach will make this manuscript much better. Some discussion is given in chapters dealing with results and conclusion but that is not enough.

  1. Please support the conclusions with the data – most important results.

---

Author Response

We would like to extend our sincerest gratitude to Reviewer 4 for your insightful and detailed review. Your comments have been invaluable in improving our work, and we sincerely appreciate the time and effort you have put in to evaluate our submission. Please find the corrected file and our response attached, and thank you again for your support.

Round 2

Reviewer 2 Report

The author's addressed all my concerns

Reviewer 3 Report

The manuscript is acceptable after revising. 

Reviewer 4 Report

The manuscript is revised according to my comments.